# Sources of skill in lake temperature, discharge and ice-off seasonal forecasting tools

François Clayer[1], Leah Jackson-Blake[1], Daniel Mercado-Bettín[2,3], Muhammed Shikhani[4], Andrew French[5†], Tadhg Moore[6‡], James Sample[1], Magnus Norling[1], Maria-Dolores Frias[7,8], Sixto Herrera[7], Elvira de Eyto[5], Eleanor Jennings[6], Karsten Rinke[4], Leon van der Linden[9], Rafael Marcé[2,3].

[1]Norwegian Institute for Water Research (NIVA), Oslo, Norway
[2]Catalan Institute for Water Research (ICRA), Girona, Spain
[3]Universitat de Girona, Girona, Spain
[4]Department of Lake Research, Helmholtz Centre for Environmental Research, Magdeburg, Germany
[5]Foras na Mara - Marine Institute, Furnace, Newport, Co. Mayo, Ireland
[6]Dundalk Institute of Technology, Dundalk, Co. Louth, Ireland
[7]Dept. Matemática Aplicada y Ciencias de la Computación (MACC). Universidad de Cantabria, Santander, Spain
[8]Grupo de Meteorología y Computación, Universidad de Cantabria, Unidad Asociada al CSIC, Santander (Spain)
[9]SA Water, Adelaide SA 5000, Australia

*Correspondence to*: François Clayer (francois.clayer@niva.no)

†Present address: School of Biological, Earth and Environmental Sciences, University College Cork, Cork T23 N63K, Ireland
‡ Present address: Department of Biological Sciences, Virginia Tech, Blacksburg, VA, USA

**Abstract.** Despite high potential benefits, the development of seasonal forecasting tools in the water sector has been slower than in other sectors. Here we assess the skill of seasonal forecasting tools for lakes and reservoirs set up at four sites in Australia and Europe. These tools consist of coupled hydrological catchment and lake models forced with seasonal meteorological forecast ensembles to provide probabilistic predictions of seasonal anomalies in water discharge, temperature and ice-off. Successful implementation requires a rigorous assessment of the tools' predictive skill and an apportionment of the predictability between legacy effects and input forcing data. To this end, models were forced with two meteorological datasets from the European Centre for Medium Range Weather Forecasts (ECMWF), the seasonal forecasts SEAS5 with three-month lead times and the ERA5 reanalysis. Historical skill was assessed by comparing both model outputs, i.e., seasonal lake hindcasts (forced with SEAS5) and pseudo-observations (forced with ERA5). The skill of the seasonal lake hindcasts was generally low although higher than the reference hindcasts, i.e., pseudo-observations, at some sites for certain combinations of season and variable. The SEAS5 meteorological predictions showed less skill than the lake hindcasts. In fact, skillful lake hindcasts identified for selected seasons and variables were not always synchronous with skillful SEAS5 meteorological hindcasts, raising questions on the source of the predictability. A set of sensitivity analyses showed that most of the forecasting skill originates from legacy effects, although during winter and spring in Norway some skill was coming from SEAS5 over the three-month target season. When SEAS5 hindcasts were skillful, additional predictive skill originates from the interaction between legacy and SEAS5 skill. We conclude that lake forecasts forced with an ensemble of boundary conditions resampled from historical meteorology are currently likely to yield higher quality forecasts in most cases.

## 1. Introduction

Freshwater provides essential services for food and energy production, manufacturing, cultural heritage, and natural habitats. However, it is threatened by more frequent extreme events (Jeppesen et al., 2021), climate change (Labrousse et al., 2020), anthropogenic water depletion (Yi et al., 2016) and agricultural pressures (Wuijts et al., 2021). Implementation of mitigation measures can help preserve freshwater resources, although they come with trade-offs between production from economic sectors with related social benefits, and availability of good quality freshwater. Hence, successful implementation of measures requires capacity at the local-regional level for cross-sectoral decision-making (Wuijts et al., 2021). Seasonal forecasting tools for water quality can help facilitate the decision-making process by informing optimal actions over the next season, e.g., magnitude and timing of reservoir drawdowns. Indeed, they can supply knowledge on the impacts of future climatic conditions on freshwater over a realistic time frame enabling implementation with reduced negative effects on economic activities. Nevertheless, the use and access to forecasting tools is still very limited for water managers (Lopez & Haines, 2017; Soares et al., 2018). The probabilistic nature of seasonal forecasts can be a key barrier coupled with the lack of reliability and credibility of these predictions in most regions outside the tropics. Hence, a better access to seasonal forecasting tools as well as increased comprehension and description of these tools are required prior to their successful implementation in the decision-making process within the water sector.

Seasonal meteorological predictions provide a probabilistic description of the weather over the next few months, e.g., an 80% chance of the weather being wetter than normal. Seasonal climate predictability mainly originates from ocean–atmosphere interactions (Troccoli, 2010). In fact, the ocean inertia, given its volume and the heat capacity of liquid water, exerts an influence on the atmosphere on the scale of months which allows us to estimate its future effect on weather. Given that ocean–atmosphere interactions are relatively strong in the equatorial region (Troccoli, 2010), seasonal meteorological predictions typically show stronger predictive skill, or prediction performance, around the tropics (Johnson et al., 2019; Manzanas et al., 2014). Under higher latitudes, skills from seasonal meteorological predictions are patchy and less consistent among variables and seasons. Hence, the boundary conditions, e.g., seasonal air temperature forecasts used to force a hydrological model, are usually not the main source of predictability outside the tropics, at least for stream flow (Greuell et al., 2019; Harrigan et al., 2018; Wood et al., 2016). Nevertheless, climate models producing seasonal meteorological forecasts are constantly improving and it is reasonable to expect that forecast opportunities will expand in the future (Mariotti et al., 2020). Developing seasonal forecasting workflows, quantifying the skill and investigating the source of predictability represent a necessary and essential step towards reliable water quality seasonal forecasting.

While some of the first forecasting tools were originally developed for flood warnings (e.g., Pagano et al., 2014; Werner et al., 2009), applications to other sectors are becoming more frequent. In the agricultural sector, for example, a recent study shows that flowering time can be reliably predicted from seasonal meteorological forecasts in central and eastern Europe, enabling early variety selection and planning of farm management (Ceglar & Toreti, 2021). Seasonal meteorological forecasts were also shown to provide useful information for the wind energy sector (Lledó et al., 2019), and to avoid significant economic losses

from hydropower generation during droughts (Portele et al., 2021). Nevertheless, the use of seasonal meteorological forecasts for water temperature in lakes and reservoirs has been limited so far, where the focus has been on water quantity (Arnal et al.,

2018; Giuliani et al., 2020; Greuell et al., 2019; Pechlivanidis et al., 2020). Studies forecasting water temperature, a fundamental water quality variable, are rare in the literature (though see Mercado-Bettin et al., 2021; Zhu et al., 2020; Baracchini et al., 2020), despite the diverse influence of this variable on lake ecosystem structure and functioning (Dokulil et al., 2021). Nevertheless, a simple lumped model (*air2water*; Piccolroaz et al., 2013), previously developed to estimate surface lake water temperature as a function of air temperature, has been applied to predict water temperature in thousands of lakes

(Zhu et al., 2021). While this hybrid approach yielded skillful surface lake water temperature forecasts (Piccolroaz et al., 2018; Toffolon et al., 2014), it doesn't allow forecasting other lake variables, such as bottom temperature.

Research on seasonal forecasting in hydrology started more than a decade ago (Troin et al., 2021) and now represents a source of knowledge for other research fields. When forecasting river flow, for example, predictability can originate from two main sources: (i) initial conditions such as catchment water stores of initial soil moisture, groundwater, and snowpack, which are

directly linked to the water residence time; and (ii) boundary conditions, i.e., meteorological forecasts used to force the hydrological model (Greuell et al., 2019). Throughout the many studies of river flow seasonal forecasting in Europe, it appears that initial conditions form the dominant source of skill in run-off (Greuell et al., 2019; Harrigan et al., 2018; Wood et al., 2016) and predictability can be extended up to a year ahead in case of very low flow as antecedent groundwater level is the key driver (Staudinger & Seibert, 2014). When dealing with standing water bodies, antecedent conditions are also likely to

provide significant predictability, given that the water storage in lakes and reservoirs is large compared to river channels, providing higher inertia. Water residence time is thus expected to exert a strong influence on discharge predictability. Water temperature, on the other hand, is influenced by multiple meteorological variables, e.g., wind, air temperature and radiation, in addition to water stores which can affect the source of its predictability.

Here, we further investigate the performance and in particular the source of this prediction performance, also referred to as

predictive or forecasting skill, of lake seasonal forecasting tools first described by Mercado-Bettin et al. (2021) and Jackson-Blake et al. (2022). These tools integrate hydrological catchment and physical lake models forced with seasonal meteorological forecasts with three-month lead times at four case study sites in Europe and Australia (Fig. 1). The meteorological variables used to force the models as well as output catchment and lake variables are a set of retrospective seasonal forecasts for past dates, hereafter referred to as hindcasts, that can be compared to historical records. The objective of this study is to assess

whether seasonal meteorological hindcast ensembles with three-month lead time, used as inputs to catchment and lake process-based models, provide some predictive skill to seasonal lake hindcasts. To this end, the forecasting skill of the tools was assessed for combinations of season and freshwater variables, i.e., discharge, water temperature or ice-off, and for each tercile. Ice-off is defined as the first ice-free day after an ice-covered period. In parallel, we quantified the forecasting skill of each meteorological variable of the seasonal meteorological prediction at each site. Both assessments were carried out following

aggregation of model outputs from daily to seasonal temporal resolution, i.e., seasonal means or sums. When a hindcast was found to perform significantly better than a reference hindcast, e.g., climatology from pseudo-observations as defined in the

Methods, for a combination of a given season, variable and tercile, this latter combination was defined as a "window of opportunity". This terminology is introduced to emphasize the fact that these forecasts can be used in the decision-making processes by water managers but only for a specific variable and season. A set of sensitivity analyses was performed to identify

input-output relationships and to partition the source of the prediction skill for each window of opportunity among warm-up, first lead-month and seasonal meteorological predictions. The comparison between hindcasts, with the aim of isolating the contributions of different sources of skill, has been applied before on streamflow hindcasts (e.g., Arnal et al., 2018; Greuell et al., 2019). However, this is, to our knowledge, the first study investigating the origin of seasonal hindcast ensemble skill on water discharge, temperature and ice-off in lakes and reservoirs. The implications for lake forecasting tools are discussed.

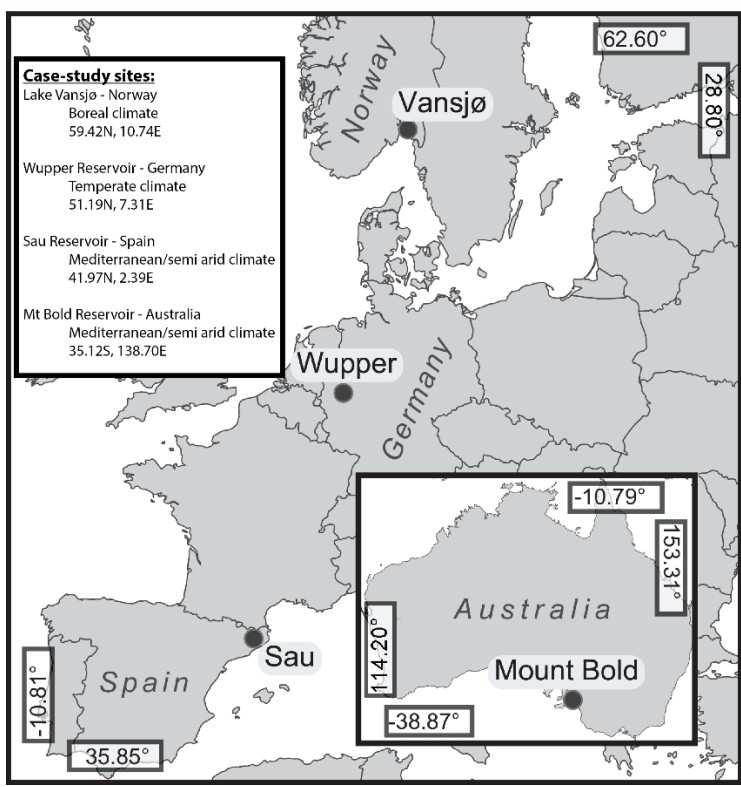


**Figure 1: Location of the four case studies in Europe and Australia along with climate type and coordinates. Map is modified from Jackson-Blake et al. (2022). Detailed catchment maps are given in Jackson-Blake (2022).**

## 2. Methods

### 2.1 Description of the forecasting tools

The forecasting tools consist of a coupled catchment runoff model to a one-dimensional water column lake model, forced by seasonal meteorological predictions, to simulate three output variables at daily resolution: inflow discharge, and lake surface and bottom temperature. For Lake Vansjø in Norway, the timing of ice melt (ice-off) was also included in the output variables in spring. The workflow consisted in running the catchment models first, providing inflow water discharge and water temperature to the lake models.

#### 2.1.1 Case study sites

Lake forecasting tools were developed for four regulated water lakes/reservoirs in Europe and Australia which have been described earlier (Mercado-Bettin et al., 2021;Table 1; Fig. 1). Briefly, Sau (Spain) and Mount Bold (Australia) reservoirs large water supplies for the cities of Barcelona and Adelaide, respectively. Lake Vansjø (Norway) is a drinking water source for three municipalities and the Wupper reservoir (Germany) is used for flood control, environmental flows, and recreation.

Table 1: Characteristics of the study sites. Mixing timing refers to boreal seasons only.

| Case study (Country) | Catchment area (km²) | Surface area (ha) | Volume (hm3) | Water retention time (yrs) | Max. Depth (m) | Mixing regime | Mixing timing |
|---|---|---|---|---|---|---|---|
| Sau (Spain) | 1680 | 575 | 165 | 0.2 | 60 | monomictic | Winter |
| Mt Bold (Australia) | 357 | 254 | 46.4 | 0.2-0.6 | 44.5 | monomictic | Summer |
| Vansjø (Norway) | 690 | 3600 | 252 | 1.1 | 19 | dimictic | Spring Fall |
| Wupper (Germany) | 215 | 211 | 26 | 0.2 | 31 | dimictic | Spring Fall |

#### 2.1.2 Meteorological input data

We used two different meteorological datasets to force the catchment hydrological and lake physical models in our tools: a climate reanalysis (ERA5) and a seasonal forecasting product (SEAS5) which both offer a global spatial and continuous temporal coverages to ensure future transferability of our workflows and easy comparison between our case-studies (Johnson

et al., 2019). ERA5 is the latest reanalysis at 0.25° spatial resolution (Hersbach et al., 2020) produced by the European Centre for Medium Range Weather Forecasts (ECMWF; https://www.ecmwf.int) within the Copernicus Climate Change Service (C3S, https://climate.copernicus.eu/). ERA5 data (1988-2016) were used (i) to correct for bias in the SEAS5 data using the quantile mapping technique as described below; (ii) to provide meteorological pseudo-observations for retrospective skill evaluation of SEAS5 hindcasts, (iii) to force catchment hydrological and lake physical models to produce pseudo-observations

of the output variables, (iv) to force our catchment and lake models to produce antecedent/warm-up period data preceding

seasonal hindcast periods (i.e., combined one lead-month and three-month target season). SEAS5 is the latest seasonal forecasting system from the ECMWF at 1° spatial resolution and provides operational seasonal forecasts and retrospective seasonal forecasts for past years (hindcasts). We used hindcasts (1994-2016) in this study. A hindcast with 25 members was considered for the period 1994-2016 for the three-month boreal seasons (spring: March through May; summer: June through

August; autumn: September through November; winter: December through February), with one month as lead time. A dedicated R package (climate4R; Iturbide et al., 2019) was used for ERA5 and SEAS5 meteorological data pre-processing. SEAS5 members were pre-processed using the quantile mapping technique (Gutiérrez et al., 2019) to correct for systematic bias relative to pseudo-observations (ERA5 reanalysis). We used the empirical quantile mapping approach (EQM) due to its ability to deal with multivariate problems (Wilcke et al., 2013). EQM adjusts 99 percentiles and linearly interpolates inside

this range every two consecutive percentiles; outside this range, a constant extrapolation (using the correction obtained for the 1st or 99th percentile) is applied (Déqué, 2007). In the case of precipitation, we applied the wet-day frequency adaptation proposed by Themeßl et al. (2011). The resulting bias-corrected data were used for hydrologic and lake models meteorological forcing, noting that we implemented bias-correction using leave-one-(year)-out cross-validation. Therefore, for each year, seasonal climate hindcast member predictions were adjusted with the bias correction parameters derived from training with all

other years; after which all bias-corrected data were appended to obtain a corrected (i.e., locally calibrated) time series of seasonal meteorological hindcasts for the full period for each case study. Finally, to use the bias-corrected data as meteorological forcing for hydrologic and lake models, we used bilinear interpolation (akima method), whereby we specified lake/reservoir coordinates from which seasonal meteorological hindcast data from surrounding pixels were interpolated. Meteorological datasets include daily average 2-meter air temperature, u and v components of wind, surface air pressure,

relative humidity (or dew-point temperature), cloud cover, short-wave radiation, downwelling long-wave radiation and daily sum of precipitation.

### 2.1.3    Observations

Daily inflow discharge and daily to monthly lake water temperature observations (Table S1) were used for catchment and lake model calibration and validation, as well as quantification of forecasting skills. For Lake Vansjø, daily measurements of

discharge over 1994–2016 were taken from the gauging station at Høgfoss (Station 3.22.0.1000.1; Norwegian Water Resources and Energy Directorate). Lake temperature data were gathered from the Vansjø-Hobøl monitoring program dataset, conducted by the Norwegian Institute for Bioeconomy Research and by the Norwegian Institute for Water Research (Haande et al., 2016). These data are available freely on the Norwegian national database (https://vannmiljo.miljodirektoratet.no).  For Sau reservoir, daily measurements of discharge into Sau Reservoir were provided by the Catalan water agency (Agència Catalana de l'Aigua,

ACA) while lake temperature and weather data are part of a long-term monitoring program (Marce et al., 2010). Discharge, water temperature and weather observations at the two other reservoir sites were collected from the water reservoir operators (Wupperverband for Wupper and SA Water for Mt Bold). Lake water temperature data are discontinuous and covered only part of the modelled time-period (1994–2016) because of limited funding for monitoring programs. In addition, precipitation,

temperature, short-wave radiation, humidity, and wind daily records at nearby meteorological stations were obtained for each
case study from the local meteorological institutes. For Lake Vansjø, this included ice-off dates from the Norwegian
Meteorological Institute station 1,715 (Rygge) located on the lake shore (59° 38′N, 10° 79′ E).

### 2.1.4 Catchment-lake process-based model setup and calibration

A catchment-lake process-based model chain was setup at each site to predict daily inflow discharge into the lake/reservoir
and daily lake water temperature. Given the specificity of each catchment regarding flow dynamics and water management,
different models were used at each site (Fig. 2). While this disparity prevents us from an in-depth comparison among case-
studies, the common methods and code established to manipulate input and output data enable us to quantify forecast
performance and the source of the predictability at each site in a consistent and comparable way.

Inflow water temperature and discharge for Sau and Vansjø was modelled with the mesoscale Hydrologic Model (mHM v5.9:
http://www.ufz.de/mhm) and SimplyQ (hydrological module of SimplyP; Jackson-Blake et al. 2017), respectively. Inflow
water temperature and discharge for Wupper and Mt Bold was modelled with the *Génie Rural* (GR) suite of models
implemented within the R package airGR (Coron et al., 2017), GR6J and GR4J, respectively. mHM and SimplyQ hydrologic
models were forced with ERA5 daily precipitation and daily average surface air temperature, and the GR models were forced
with daily precipitation and daily potential evapotranspiration (Hargreaves-Samani potential evapotranspiration, derived from
daily minimum and maximum temperature, implemented in drought4R; Iturbide et al., 2019). All hydrological models were
calibrated and validated against local observations using the Nash–Sutcliffe efficiency coefficient (NSE) as the objective
function.

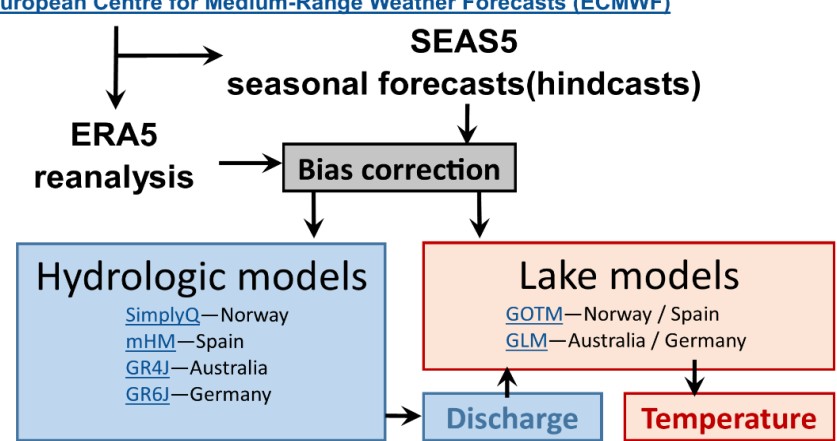

**Figure 2: Description of the forecasting workflow—Calibrated hydrologic and lake models are used to produce seasonal lake hindcasts with 25 members**

The General Ocean Turbulence Model (GOTM, http://gotm.net) was used to simulate the water temperature profile of Sau
Reservoir and Lake Vansjø. The General Lake Model (GLM, Hipsey et al., 2019) was used to simulate water temperature in

the Mt. Bold and Wupper reservoirs. Lake models were forced with ERA5 surface air temperature, u and v wind components, surface air pressure, relative humidity (or dew-point temperature), cloud cover, short-wave radiation, precipitation and, in some cases, also downwelling long-wave radiation, and calibrated and validated against observations using the Root-Mean-Square Error (RMSE) and NSE as objective functions.

For Lake Vansjø, the water level was set to constant given that observed fluctuations are < 1 m which are not critical for the lake heat and water budgets. The three reservoirs, on the other hand, experience much larger water level fluctuations because of complex water pumping patterns and/or water scarcity. It was thus critical to allow for water level fluctuations and parametrize the outflows to avoid dry outs. For Wupper Reservoir, a statistical model was developed to calculate the reservoir's outflow based on the inflow using the timeseries over the warm-up period for each discharge simulation of the catchment model. Such an approach allowed mimicking the outflow decision and approximately resembling the observed water-level to avoid the cases of dry-outs or exceedingly low volumes of water due to inflow/outflow misestimation. More details on the performance of the linear regression are given in the supplementary information. For Sau, historical observations of outflow and pumping volumes were used to force the model. For Mt. Bold Reservoir, an average annual cycle was calculated from historical observations and then replicated throughout the entire timeseries. While this assumption does not allow for inter-annual variation, it allowed for simulation of water level fluctuation each year that represented the seasonal cycle apparent within Mt. Bold and avoided dry outs.

The lake energy budget includes exchanges through the air-water interface, i.e., downward short-wave radiation, downward and upward long-wave radiation, latent and sensible heat fluxes, and by lateral fluxes of water, i.e., inflow and outflow of water (Schmid and Read 2022). The energy fluxes at the air-water interface are accounted for in the GLM or GOTM lake model, however, the lateral fluxes caused by throughflow (inflow-outflow balance) need to be parametrized through the addition of water temperature to the inflow provided by the catchment model. Inflow temperature was estimated based on the assumption that water temperatures follow the air temperatures closely with some time lag (Stefan & Preud'homme 1993; Ducharne 2008). Hence, water temperature was predicted with a linear model of the form A + B*AirTemperature where A and B were optimized against local observations when available. At Sau Reservoir, the values of A and B were 5.12 and 0.799, respectively, while for Mt Bold reservoir and Lake Vansjø, the values of A and B were 5 and 0.75, respectively. The validation of this model for Wupper Reservoir, as an example, is described in the supplementary information (SI).

Most common verification statistics, e.g., Kling-Gupta efficiency (KGE), NSE and RMSE, for hydrological and lake modeling were calculated. Details on calibration and validation periods as well as statistics are shown in Table 4 and Table S2.

### 2.1.5 Pseudo-observations (Lake_PO)

Following calibration, lake and hydrologic models were forced with ERA5 over 1994-2016 to produce daily pseudo-observations of river discharge, daily surface and bottom temperature, as well as presence or absence of ice (for Lake Vansjø only). The output of this simulation is hereafter referred to as lake pseudo-observations (Lake_PO). Theoretical prediction skill of seasonal forecasts is commonly evaluated against pseudo-observations (Greuell et al., 2019; Harrigan et al., 2018;

Wood et al., 2016). In contrast to lake real observations, Lake_PO have the advantages of being complete and allow to disregard changes in skill related to model errors or biases (Harrigan et al., 2018), and to focus on skill originating from initial and boundary conditions. In contrast to the theoretical prediction skill, the total prediction skill includes any error or bias introduced by the model. Here, the total prediction skill of seasonal lake hindcasts (discharge, water temperature and ice-off) was also evaluated against real observations, when those were available and covering a representative time period.

### 2.1.6    Seasonal forecasts (Lake_F)

For each of the 92 three-month hindcast seasons (11/1993 to 11/2016), we simulated ensemble predictions of daily river discharge, daily surface and bottom water temperature as well as presence or absence of ice (for Lake Vansjø only; Fig. 3). Catchment and lake models were forced with ERA5 data over the 1-year warm-up period followed by a set of 25 members of SEAS5 data covering the first lead month (M0) and the 3-month long target season (M1–M3). The first lead month is defined
in agreement with Greuell et al. (2019) as the month following the date on which the forecast would have been issued. Over the first lead month, the 25 members of SEAS5 progressively diverge from ERA5 to their respective SEAS5 member. Model outputs for the final 3 months, i.e., the target season, were aggregated into three month (M1–M3) seasonal averages or sums (i.e., average surface and bottom water temperature and cumulative seasonal inflow discharge). The output of this simulation is hereafter referred to as lake forecasts (Lake_F).

### 2.2    Assessment of modeling performance and source of forecasting skills

#### 2.2.1    Model and forecast verification

A complete assessment of the modelling and forecasting performance of our workflow was performed through several verifications (Table 2). The first verification (Verification 1 in Table 2) consisted in evaluating the performance of the models forced with ERA5 by comparing model outputs (Lake_PO) to observations at daily temporal resolution, as described in section
2.1.4. This verification step included the reporting of traditional verification statistics for modelling, i.e., NSE; KGE and RMSE. The second and third verifications (Verifications 2 and 3 in Table 2) consisted in quantifying the lake forecast (Lake_F) performance compared to climatology from pseudo-observations (Lake_PO) or from observations, respectively. These steps allowed to quantify the forecasting skill of a perfect model and the total forecasting skill, respectively. Forecast verification 2 and 3 were performed using model output data at seasonal temporal resolution, i.e., daily model outputs over the target season (M1–M3), were aggregated in seasonal averages or sums. For forecast verification 2 and 3, hindcast predictions are categorized
into three terciles, where the upper tercile includes data points falling in the percentile range 66–100%, the middle tercile includes data in the range 33–66%, and lower tercile includes data in the range 0–33%.

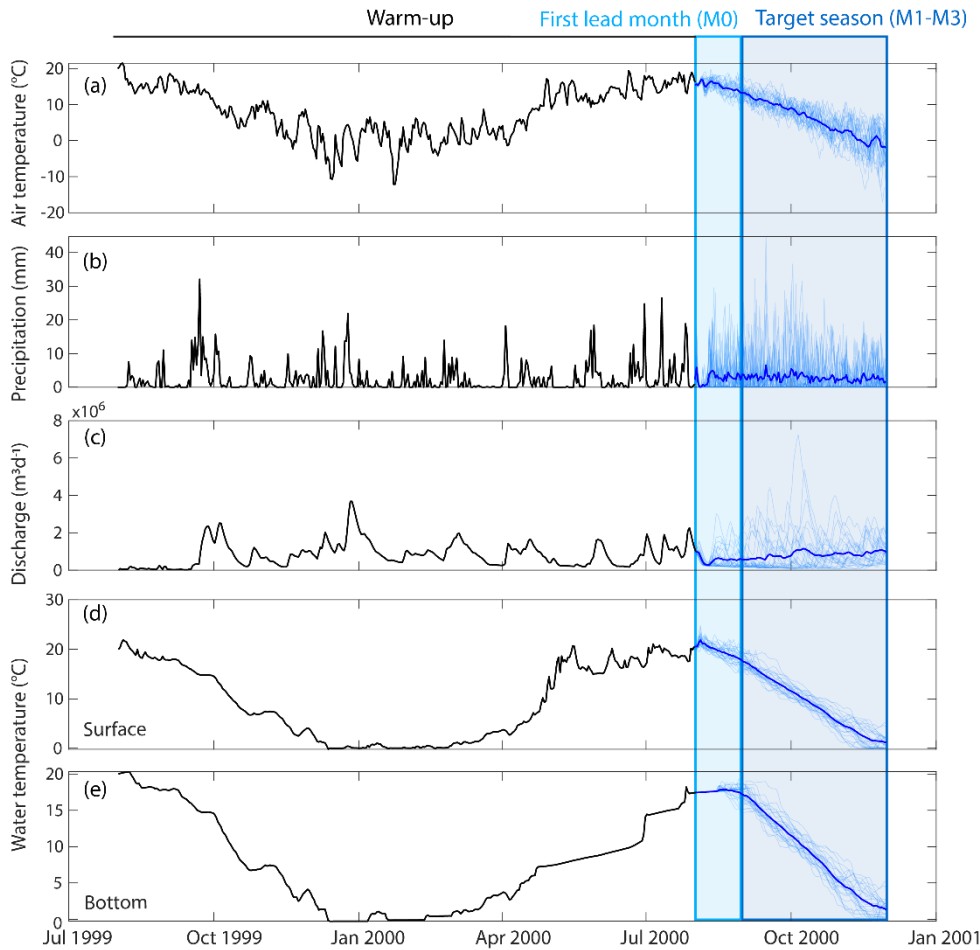

**Figure 3: Time series of the air temperature (a), precipitation (b), discharge (c), surface (d) and bottom (e) water temperature over the warm-up, first lead month (M0) and target season (M1–M3) for Autumn 2000. The black lines indicate ERA5 (a and b) and Lake_PO (c–e) data, the light and dark blue lines are, respectively, the 25 members and the mean of SEAS5 (a and b) and Lake_F (c–e).**

Forecast performance was quantified with two skill scores: the Ranked Probability Skill Score (RPSS) and the Relative Operating Characteristic Skill Score (ROCSS). Skill scores are a measure of the relative improvement of the forecast compared to a reference forecast which here is the climatology based on either Lake_PO or observations. ROCSS values were calculated against climatology from real observations ($ROCSS_{Obs}$), in addition to pseudo-observations ($ROCSS_{original}$), only when observations covered the whole season. Indeed, $ROCSS_{original}$ was calculated only if there was at least one observation point in each month of the season and observations for at least 70% of the seasons. Observations that met these criteria only included inflow discharge at Vansjø, Sau and Wupper for all seasons, surface and bottom temperature at Vansjø in summer only, surface and bottom temperature at Wupper for all seasons, surface temperature at Sau for all seasons and ice-off at Vansjø.

RPSS and ROCSS are commonly used as evaluation measures of probabilistic forecasting skill (Jolliffe & Stephenson, 2012; Müller et al., 2005). The visualizeR package (Frías et al., 2018) was used to compute the RPSS and ROCSS for Lake_PO and Lake_F. Briefly, the RPSS provides a relative performance measure on how well the probabilistic ensemble is distributed over the lower, middle and upper terciles, while the ROCSS provides a relative measure of discriminative skill for each category. A RPSS > 0 is associated with a better forecast than the reference (1 being a perfect score), while RPSS ≤ 0 indicates no improvement compared to the reference. The ROCSS value ranges from -1 (perfectly bad forecast) to 1 (perfect forecast) and a zero value indicates no skill compared to the reference. The RPSS has been shown to be sensitive to the ensemble size, but this effect can be corrected for using the Fair (or unbiased) RPSS (Ferro et al., 2014). To allow for comparison with other forecasting systems, we have used the fair RPSS (FRPSS) forecast verification. In this study, the FRPSS is calculated for tercile events. The statistical significance of the FRPSS and ROCSS is computed based on the 95% confidence level from a one-tailed Z test. When a forecast for a given season, variable and tercile was associated with a ROCSS value that was statistically significant, we referred to it as a window of opportunity (i.e., a combination of season, variable and tercile for which forecast performance was significantly better than the reference). In our case, the threshold values above which a ROCSS was considered significant typically range between 0.47 and 0.55.

**Table 2: Comparison carried out to evaluate model and forecast performance**

| Verification | Outputs used | Reference forecast data | Purpose | Statistics |
|---|---|---|---|---|
| 1 | Lake_PO | Observations | Assess lake model skill | KGE NSE RMSE |
| 2 | Lake_F | Lake_PO | Assess the transfer of meteorological forecast skill through process-based models – Perfect model forecasting skill | $ROCSS_{original}$ |
| 3 | Lake_F | Observations | Assess total forecasting skill | $ROCSS_{Obs}$ |

### 2.2.2    Sensitivity analyses to initial conditions and meteorological forcing

Several sensitivity analyses (SA), summarized in Table 3, were performed to identify the origin of the forecasting skill for a given window of opportunity, i.e., a combination of season, variable and tercile for which forecast performance was significantly better than the reference. Results of the SA are only reported for sites having a substantial number of windows of opportunity for conciseness. These SA allowed quantifying the sensitivity of hindcast performance to forcing data over specific periods: the target season (M1–M3; SEAS5), the first lead month (M0) and the warm-up period (ERA5). It was thus possible to quantify the proportion of skills originating from each of these periods.

The SA consisted of replacing the forcing data of interest, i.e., over the target season, the first lead month or the warm-up period, by data from an equivalent season/period but from a randomly selected year. For example, for the target season SA (S-

SA), the SEAS5 forcing data covering the 3-month target season was replaced by SEAS5 data from a randomly selected equivalent season. Furthermore, the SA for the warm-up period (W-SA) consisted in replacing the ERA5 data covering the warm-up period by ERA5 data from a randomly selected equivalent time-period. The last SA covered warm-up and first lead

month (W+M0-SA) and consisted in replacing ERA5 data over the warm-up, as in W-SA, but also SEAS5 data over the first lead month. To ensure that the randomly sampled forcing data are representative of the whole SEAS5 or ERA5 datasets, we introduce two levels of repetitions for all experiments. First, we randomly selected a year for each of the 25 members of SEAS5, meaning that the data selected to replace the original SEAS5 forcing data is extremely likely to be from a different year for each SEAS5 member. Second, we repeated the analysis 25 times, for each season. Sensitivity analyses were only

carried out for Spain and Norway because of the low number of windows of opportunity at the two other sites and considering the resources needed to execute these hindcast experiments.

The outputs of each of the sensitivity analysis were used to calculate ROCSS values against the climatology based on Lake_PO, as for Lake_F in the Verification 2 described above (Table 2). The ROCSS values obtained through this procedure were, respectively, $ROCSS_S$, $ROCSS_W$ and $ROCSS_{w+M0}$ for S-SA, W-SA, and W+M0-SA. The ROCSS values obtained for the

various SAs were compared to the original Lake_F ROCSS values ($ROCSS_{original}$) to investigate the sources of prediction skill. An estimation of the proportion of prediction skill originating from the SEAS5 data over the target season ($P_{season}$) was expressed as follows:

$$P_{season} = ROCSS_{original} - ROCSS_S \qquad (1)$$

Similarly, the proportions of prediction skill originating from the ERA5 data over the warm-up ($P_{warm-up}$) and from the

SEAS5 data over the first lead month ($P_{M0}$) can be respectively estimated as:

$$P_{warm-up} = ROCSS_{original} - ROCSS_W \qquad (2)$$

$$P_{M0} = ROCSS_W - ROCSS_{W+M0} \qquad (3)$$

In Eq. 1–3, prediction skill was assumed to linearly scale with ROCSS values and skill from any interaction effect was neglected. While we admit that Eqs. 1–3 are not necessarily statistically correct, they are useful to quantify the relative

importance of the sources of skill. Hence, the values of $P_{season}$, $P_{warm-up}$ and $P_{transition}$ should be interpreted with care.

### 2.2.3 Sensitivity analyses to individual input variables

To further investigate through which process forecasting skill is transferred from input to output variables, a one-at-a-time sensitivity analysis (OAT-SA) was performed for Lake_PO and the Pearson partial correlation coefficients (PPCC) between each variable of Lake_PO, i.e., surface temperature, bottom temperature, discharge, ice-off, and a set of relevant input variables

were determined (Table 3). The OAT-SA consisted in replacing the data for a specific input meteorological variable by data from an equivalent target season but from a randomly selected year. The seasonal means of OAT-SA outputs were compared to default outputs (Lake_PO) with the square of the Pearson correlation coefficient ($R^2$). Higher ($1 - R^2$) values indicate more influence of input variables on Lake_PO.

PPCC allowed quantifying the sensitivity of model outputs to a given input variable while removing the effect of the remaining input variables. Note that PPCC were calculated on seasonally aggregated variables. To ensure that PPCC were statistically appropriate, i.e., only when a linear relationship exists between the seasonal means of input factors and those of the output (Pianosi et al., 2016), the linearity assumption was checked through visual inspection of scatter plots between each input and output variables. Partial correlation coefficients are a good alternative to 'All-At-a-Time' (or global) SA when the latter is not possible because of the lack of computing resources (Pianosi et al., 2016). To avoid misleading conclusions, correlation between input variables should be minimized (Marino et al., 2008). Hence, only the most relevant input variables were included. Precipitation and air temperature were retained for discharge, while air temperature, precipitation, wind speed (wind speed calculated from u and v components of wind) and short-wave radiation were retained for surface and bottom temperature. In fact, short-wave radiation was retained over relative humidity, cloud cover and air pressure because it was responsible for most of air-water heat fluxes (see SI). Wind was retained because of its impact on thermal stability (Blottiere, 2015).

**Table 3: List of sensitivity analyses (SA) performed**

| SA | Forcing data to be replaced | | Model output | Purpose | Sensitivity index |
|---|---|---|---|---|---|
| | Period | Variable | | | |
| S-SA | Target season (SEAS5) | All | Lake_F | Quantifying the proportion of forecasting skill originating from SEAS5 data over the target season | $ROCSS_S$ |
| W-SA | Warm-up period (ERA5) | All | Lake_F | Quantifying the proportion of forecasting skill originating from ERA5 data over the warm-up season – initial conditions | $ROCSS_W$ |
| W+M0-SA | Warm-up period (ERA5) and first lead month (SEAS5) | All | Lake_F | Quantifying the proportion of forecasting skill originating from SEAS5 data over the first lead month | $ROCSS_{W+M0}$ |
| OAT-SA | Target season (ERA5) | One at a time | Lake_PO | Quantifying the sensitivity of Lake_PO to a specific forcing variable | $1 - R^2$ |
| PPCC | None | None | Lake_PO | Quantifying the sensitivity of Lake_PO to a specific forcing variable while removing the effect of the remaining variables | PPCC |

## 3.      Results

### 3.1 Performance of the calibrated catchment and lake models (Lake_PO)

Catchment and lake models calibrated against local observations performed reasonably well (Table 4). For river discharge, NSE and KGE both ranged between 0.51 and 0.85 over the calibration and validation periods. For surface water temperature,

RMSE ranged from 1.10 to 1.63 and NSE from 0.78 to 0.94 over the calibration and validation periods. Over each season, however, Lake_PO showed more heterogeneous performance (Table S2). Discharge simulations were usually worse in summer, except in Australia where performance was poor for most seasons. Surface water temperature modeling typically showed better performance during spring and fall than during summer or winter. There is no clear pattern for bottom water temperature, but overall, it seems more difficult to be accurately simulated compared to surface temperature.


**Table 4: Verification statistics of the catchment and lake model for each case study**

| | Output variable | Time | Calibration NSE | KGE | RMSE | Time | Validation NSE | KGE | RMSE |
|---|---|---|---|---|---|---|---|---|---|
| Norway | Discharge | 2005–2010 | 0.51 | 0.56 | | 2011–2015 | 0.57 | 0.57 | |
| | Temperature | 2005–2010 | 0.92 | | 1.12 | 2011–2015 | 0.93 | | 1.10 |
| Spain | Discharge | 1997–2007 | 0.60 | 0.66 | | 2008–2018 | 0.54 | 0.63 | |
| | Temperature | 1997–2007 | 0.93 | | 1.63 | 2008–2018 | 0.94 | | 1.45 |
| Germany | Discharge | 1991–2011 | 0.71 | 0.85 | | 2012–2016 | 0.63 | 0.81 | |
| | Temperature | 1993–2010 | 0.93 | | 1.31 | 2011–2016 | 0.91 | | 1.53 |
| Australia | Discharge | 2003–2007 | 0.64–0.80 | 0.70–0.84 | | 2008–2013 | 0.65–0.80 | 0.54–0.75 | |
| | Temperature | 2014–2016 | 0.91 | | 1.17 | 2016–2018 | 0.78 | | 1.50 |

### 3.2 Skill of the seasonal meteorological (SEAS5) and Lake (Lake_F) hindcasts

Table 5 displays the ROCSS values for each combination of Lake_F output variable, season and tercile while Table 6
summarizes the windows of opportunity, i.e., a combination of season, variable and tercile for which forecast performance, or predictive skill, was significantly better than the reference, for SEAS seasonal meteorological hindcasts as well as for Lake_F hindcasts. These windows of opportunity typically had ROCSS values larger than 0.47 to 0.55 (see Methods section for details). For SEAS5 seasonal meteorological hindcasts, only 3 to 10 windows of opportunity were observed for each case study out of the 96 possibilities, i.e., 3 terciles of 8 variables over 4 seasons (Table 6). Regarding Lake_F, larger proportions of the 36–39
possible variable-tercile-season combinations were associated with statistically significant ROCSS values (Table 6). Winter and Spring in Norway, as well as Summer and Autumn in Spain were the seasons associated with the most skillful Lake_F hindcasts. Lake Vansjø in Norway was the only case study where windows of opportunity for SEAS5 and Lake_F were consistently concentrated within the same seasons, i.e., mostly in Spring and to a lesser extent in Winter. For the other case studies, there were fewer windows of opportunity for SEAS5 and those were more randomly distributed over the year. FRPSS
values were typically reported for surface water temperature in spring and autumn, except for autumn in Spain. Norway and Germany also showed significant fair RPSS for bottom water temperature in spring and autumn, and summer and autumn, respectively. Note that neither river discharge nor any of the SEAS5 variables had FRPSS values in any case study. Windows

of opportunity for bottom temperature represented more than half of the total for all case-studies and variables while those for surface temperature and discharge were more sporadic.

The comparison of SEAS5 and Lake_F skillful hindcasts in Table 6 is already useful for identifying possible transfer of forecasting skill from the SEAS5 seasonal meteorological hindcasts to the catchment and lake models. SEAS5 meteorological hindcasts are skillful over only a very limited number of seasons, variables and terciles (Table 6). However, for Norway, there is a higher number of skillful meteorological and lake hindcasts in spring than in the other seasons. For the other case-studies, such a clear connection between SEAS5 meteorological hindcasts and catchment/lake model outputs is not as apparent. We

can thus hypothesize that the skill of catchment and lake model hindcasts in Norway is more inherited from the SEAS5 data than at other case studies. In contrast, skill of the catchment and lake model hindcasts at the other case-studies is hypothesized to originate from the legacy of the warm-up period or from the parametrization of the inflow-outflow water balance.

Verification statistics for Lake_PO seasonal means compared to observations (Table 7) show that the catchment and lake models performed well at the Norwegian and Spanish sites in capturing interannual variability. In Germany and Australia,

performance was lower. Note that when observation coverage was below 50%, no statistics were calculated given the low number of seasons represented and the risk of bias when computing seasonal averages. The difference between $ROCSS_{original}$ .(comparing Lake_F and Lake_PO) and $ROCSS_{Obs}$ (comparing Lake_F and lake observations) did not necessarily scale inversely with the verification statistics (Table 7). In fact, the $ROCSS_{Obs}$ reported for the German site were slightly lower or even larger than their respective $ROCSS_{original}$ with differences lower than 0.23. Whereas, for the Spanish

site, three $ROCSS_{Obs}$ values out of 4 were significantly lower than the $ROCSS_{original}$ with a difference larger than 0.33. Nevertheless, several output variables, e.g., bottom temperature in Germany and ice-off in Norway, are associated with significant $ROCSS_{original}$ .and $ROCSS_{Obs}$ .which provides further confidence in model calibration and low model error. In contrast, even if the verification statistics for discharge were not worse than for the other variables, $ROCSS_{Obs}$ values are all below the significance threshold pointing towards some limitations in predicting hydrology.

**Table 5:** $ROCSS_{original}$ for each combination of season, variable and tercile of lake hindcasts (Lake_F). Color scale range from dark blue (ROCSS = -1, perfectly bad forecast) to dark red (ROCSS = 1, perfect forecast) with white in the middle (ROCSS = 0, no change compared to reference forecast). Windows of opportunity are highlighted by bold, black numbers, i.e., combination of season, variable and tercile associated with a statistically significant ROCSS value.

| | | Norway | | | Spain | | | Germany | | | Australia | | |
|---|---|---|---|---|---|---|---|---|---|---|---|---|---|
| | | Lower | Middle | Upper | Lower | Middle | Upper | Lower | Middle | Upper | Lower | Middle | Upper |
| Discharge | Spring | **0.58** | 0.18 | **0.54** | 0.37 | -0.11 | 0.33 | -0.34 | -0.03 | -0.47 | 0.01 | -0.33 | -0.64 |
| | Summer | -0.13 | 0.23 | -0.41 | **0.73** | -0.02 | 0.23 | -0.59 | -0.13 | -0.34 | -0.05 | 0.28 | 0.36 |
| | Autumn | 0.32 | -0.24 | 0.27 | 0.17 | 0.46 | **0.47** | -0.29 | -0.33 | 0.03 | **0.48** | -0.02 | 0.18 |
| | Winter | 0.21 | -0.08 | -0.1 | 0.4 | **0.52** | 0.08 | -0.28 | 0.17 | -0.34 | 0.16 | -0.02 | -0.22 |
| Surface Temperature | Spring | **0.75** | 0.14 | **0.53** | 0.2 | 0.18 | 0.19 | 0.2 | -0.14 | 0.38 | 0.19 | -0.27 | 0.26 |
| | Summer | 0.33 | 0.39 | -0.13 | 0.42 | -0.13 | **0.57** | -0.25 | -0.18 | -0.22 | 0.16 | -0.48 | 0.23 |
| | Autumn | -0.52 | -0.24 | -0.12 | 0.22 | 0.11 | 0.47 | 0.23 | 0.42 | 0.3 | 0.13 | -0.34 | -0.04 |
| | Winter | **0.48** | -0.12 | -0.17 | 0.32 | -0.66 | -0.15 | -0.22 | 0.04 | -0.09 | 0.13 | -0.14 | -0.1 |
| Bottom Temperature | Spring | **0.56** | 0.05 | **0.68** | 0.44 | 0.12 | **0.86** | **0.59** | 0.28 | **0.6** | 0.37 | 0.31 | -0.16 |
| | Summer | -0.12 | 0.14 | -0.35 | **0.53** | 0.36 | **0.72** | **0.48** | 0.21 | **0.71** | **0.6** | -0.45 | 0.18 |
| | Autumn | -0.53 | 0.03 | -0.26 | **0.5** | 0.55 | **0.64** | 0.27 | 0.26 | 0.15 | 0.28 | 0.4 | 0.38 |
| | Winter | **0.48** | 0.01 | **0.53** | -0.02 | 0.54 | 0.27 | -0.16 | -0.3 | 0.27 | **0.63** | -0.1 | 0.29 |
| Ice-off | Spring | **0.69** | 0.29 | **0.75** | | | | | | | | | |

**Table 6:** SEAS5 meteorological and Lake_F lake hindcasts associated with statistically significant FRPSS or ROCSS at each case-study.

| Site | Indexes | Winter | | Spring | | Summer | | Autumn | | TOTAL | |
|---|---|---|---|---|---|---|---|---|---|---|---|
| | | SEAS5 | Lake_F | SEAS5 | Lake_F | SEAS5 | Lake_F | SEAS5 | Lake_F | SEAS5 | Lake_F |
| Norway | FRPSS | | | | ST; BT | | | | ST; BT | 0/32 | **4/12** |
| | ROCSS | 3 cc (−) sw (+) lw (=) | 3 ST (−) BT (−,+) | 7 airP(=,+) airT (+) cc (=) hum (−) U (+) V (−) | 8 Q (−,+) ST (−,+) BT (−,+) Ice-off (−,+) | 0 | 0 | 0 | 0 | **10/96** | **11/39** |
| Australia | FRPSS | | | | ST | | | | ST | 0/32 | **2/12** |
| | ROCSS | 2 airP (+) hum (+) | 1 BT (−) | 1 lw (+) | | 1 cc (=) | 1 BT (−) | 4 airP (+) cc (+) airT (=) sw (+) | 1 Q (−) | **8/96** | **3/36** |
| Spain | FRPSS | | | | ST | | | | | 0/32 | **1/12** |
| | ROCSS | 0 | 1 Q (=) | 2 cc (+) airP (+) | 1 BT (+) | 2 cc (+) hum (+) | 5 Q (−) ST (+) BT (−,+) | 1 cc (+) | 3 Q (+) BT (−,+) | **5/96** | **9/36** |
| Germany | FRPSS | | | | ST | | BT | | ST; BT | 0/32 | **4/12** |
| | ROCSS | 2 lw (=) V (−) | 0 | 1 hum (+) | 2 BT (−,+) | 0 | 2 BT (−,+) | 0 | 0 | **3/96** | **4/36** |

Lake_F variable abbreviations: ST, BT and Q stand for surface-, bottom-temperature and discharge, respectively. SEAS5 meteorological variable abbreviations: airT, airP, cc, hum, sw, lw, U, V stand for surface air temperature, air pressure, cloud cover, relative humidity (or dew-point temperature), short-wave radiation, downwelling long-wave radiation, and u and v components of wind, respectively. -, + and = stands for lower, upper and middle terciles, respectively.

**Table 7: Verification statistics (NSE, R², RMSE, RMSE/sd, bias) for Lake_PO seasonal means (comparing Lake_PO to observations), as well as comparison of the $ROCSS_{original}$ (comparing Lake_F and Lake_PO) and $ROCSS_{Obs}$ (comparing Lake_F and lake observations).**

| Site | Variable | Season | Obs coverage S | M | D | NSE | R² | RMSE | RMSE/sd | bias | $ROCSS_{original}$ lower | middle | upper | $ROCSS_{Obs}$ lower | middle | upper |
|---|---|---|---|---|---|---|---|---|---|---|---|---|---|---|---|---|
| Norway | Discharge | Spring | 100 | 96 | 93 | 0.72 | 0.80 | 2.0 | 0.52 | -1.0 | 0.58* | | 0.54* | 0.36 | | 0.34 |
| | Surface | Winter | 0 | 0 | 0 | | | | | | 0.48* | | | | n.a | |
| | Temperature | Spring | 48 | 58 | 5 | | | | | | 0.75* | | 0.53* | n.a | | n.a |
| | Bottom | Winter | 0 | 0 | 0 | | | | | | 0.48* | | 0.53* | n.a | | n.a |
| | Temperature | Spring | 43 | 52 | 4 | | | | | | 0.56* | | 0.68* | n.a | | n.a |
| | Ice-on | | 100 | - | - | 0.97 | 0.99 | 2.2 | 0.16 | 1.8 | | | a | | | |
| | Ice-off | | 100 | - | - | 0.36 | 0.76 | 19.3 | 1.09 | -14.7 | 0.69* | | 0.75* | 0.55* | | 0.68* |
| Spain | Discharge | Winter | 100 | 100 | 99 | 0.88 | 0.89 | 3.9 | 0.34 | -0.6 | | 0.52* | | | 0.18 | |
| | | Summer | 100 | 100 | 98 | 0.51 | 0.62 | 3.5 | 0.69 | -1.6 | 0.73* | | | 0.40 | | |
| | | Autumn | 100 | 100 | 98 | 0.73 | 0.74 | 4.0 | 0.51 | -0.8 | | | 0.47* | | | 0.40 |
| | Surf. Temp. | Summer | 78 | 78 | 3 | 0.12 | 0.40 | 1.1 | 0.87 | -0.6 | | | 0.57* | | | -0.08 |
| | Bottom | Spring | 48 | 70 | 3 | | | | | | | | 0.86* | | | n.a |
| | Temperature | Summer | 48 | 67 | 2 | | | | | | 0.53* | | 0.72* | n.a | | n.a |
| | | Autumn | 35 | 58 | 3 | | | | | | 0.50* | | 0.64* | n.a | | n.a |
| Ger. | Bottom | Spring | 100 | 96 | 6 | -5.01 | 0.49 | 1.2 | 2.40 | 1.0 | 0.59* | 0.60* | | 0.41 | | 0.46* |
| | Temperature | Summer | 100 | 100 | 7 | -8.63 | 0.26 | 3.8 | 3.04 | 3.6 | 0.48* | | 0.71* | 0.51* | | 0.49* |
| Australia | Discharge | Autumn | 43 | 100 | 100 | -0.67 | 0.41 | 1.61 | 1.23 | -1.27 | 0.48* | | | | n.a | |
| | Bottom | Winter | 23 | 100 | 82 | -0.70 | 0.32 | 1.98 | 1.17 | 1.51 | 0.63* | | | | n.a | |
| | Temperature | Summer | 17 | 75 | 46 | | | | | | 0.60* | | | | n.a | |

Only output variables associated with statistically significant ROCSS_original are included. Statistically significant ROCSS are highlighted with an asterisk.

"Obs coverage" is the percentage of seasons (S), months (M) and days (D) covered by observations. Spring is March to May, Summer is June to August, Autumn is September to November, and Winter is December to February.

[a]Ice-on typically occurs between November and December which is the autumn and winter boundary. Therefore, ROCSS values could not be calculated for ice-on.

### 3.3 Sensitivity analyses to initial conditions and meteorological forcing

The $ROCSS_S$, $ROCSS_W$ and $ROCSS_{W+M0}$ values obtained for each run of S-SA, W-SA and W+M0-SA, respectively, are summarized in boxplots in Figure 4 together with the original ROCSS value for each window of opportunity at the Norwegian and Spanish sites. This set of sensitivity analyses (SA) were performed to identify the origin of the forecasting skill for a given window of opportunity and allowed quantifying the sensitivity of hindcasts performance to forcing data over specific periods: the target season (M1–M3; SEAS5), the first lead month (M0) and the warm-up period (ERA5). In general, output variable

sensitivity to SEAS5 data over the target season (S-SA) is small relative to sensitivity to ERA5 data over the warm-up season and/or SEAS5 data over the first lead month. In fact, at Sau, replacing SEAS5 data over the target season with random data (S-SA) does not yield any significant change in the ROCSS values, except for the surface temperature upper tercile (Fig. 4 panel l). However, significant changes in ROCSS values are seen for W-SA compared to $ROCSS_{original}$ indicating high sensitivity to warm-up. The similar ranges in $ROCSS_W$ and $ROCSS_{W+M0}$ values suggest limited or no impact of the SEAS5 data over the first lead month on output variable forecasts.

At Vansjø in Norway, on the other hand, 8 out of 11 windows of opportunity show significant changes in $ROCSS_S$ values, indicating higher sensitivity to SEAS5 data over the target season than at Sau. Furthermore, 3 windows of opportunity are associated with $ROCSS_S$ that are lower than $ROCSS_{original}$ (Fig. 4b, f and g), i.e., suggesting SEAS5 is providing some skill, while 5 have $ROCSS_S$ that are higher than $ROCSS_{original}$ (Fig. 4a, h–k), suggesting the use of SEAS5 is in fact reducing forecasting skill compared to a random forecast. Then, a progressive decrease in ROCSS values is typically observed for all windows of opportunity following W-SA and W+M0-SA, indicating a progressive loss of forecasting skill related to ERA5 data over the warm-up and SEAS5 data over the first lead month.

### 3.4 Sensitivity analyses to specific input variables

Figure 5 and 6 summarizes the results from the two sensitivity analyses to specific input variables: OAT-SA and PPCC. Seasonal means of Lake_PO at Vansjø also showed higher sensitivity to specific input variables than Lake_PO at Sau (Fig. 5). In fact, surface temperature is highly sensitive to surface air temperature over the year while some other input variables have more specific influence. Bottom temperature is also highly sensitive to surface air temperature but wind also plays a large role, especially in summer which is consistent with its expected impact on lake thermal stability (Blottiere, 2015). Finally, as expected, discharge at Vansjø is highly sensitive to precipitation, and to a lesser degree to surface air temperature, except in winter where surface air temperature has a larger influence on discharge.

The PPCC also show similar patterns regarding sensitivity (Fig. 6) where discharge is highly correlated with precipitation at the four sites and surface air temperature plays a secondary role for specific seasons. Once again, surface and bottom temperature at Sau stand out due to their limited sensitivity to input variables while at the three other sites, surface temperature, and to a lesser degree bottom temperature, are generally strongly positively correlated with surface air temperature. Others, like precipitation and short-wave radiation have more of an anecdotal influence on lake temperature, while wind shows a more consistent negative impact on surface temperature at Vansjø, Wupper and Mt Bold. Wind also shows some impact on bottom temperature, although less consistent. At Vansjø and Mt Bold following the coldest season, wind is positively correlated with bottom temperature, while at Wupper during the two coldest seasons, wind is negatively correlated with bottom temperature. Finally, ice-off date in Vansjø shows a strong negative correlation with surface air temperature (Fig. 6m) that can be linked back to the snow content and the intensity of snow melt in the catchment (Fig. 6n and o).

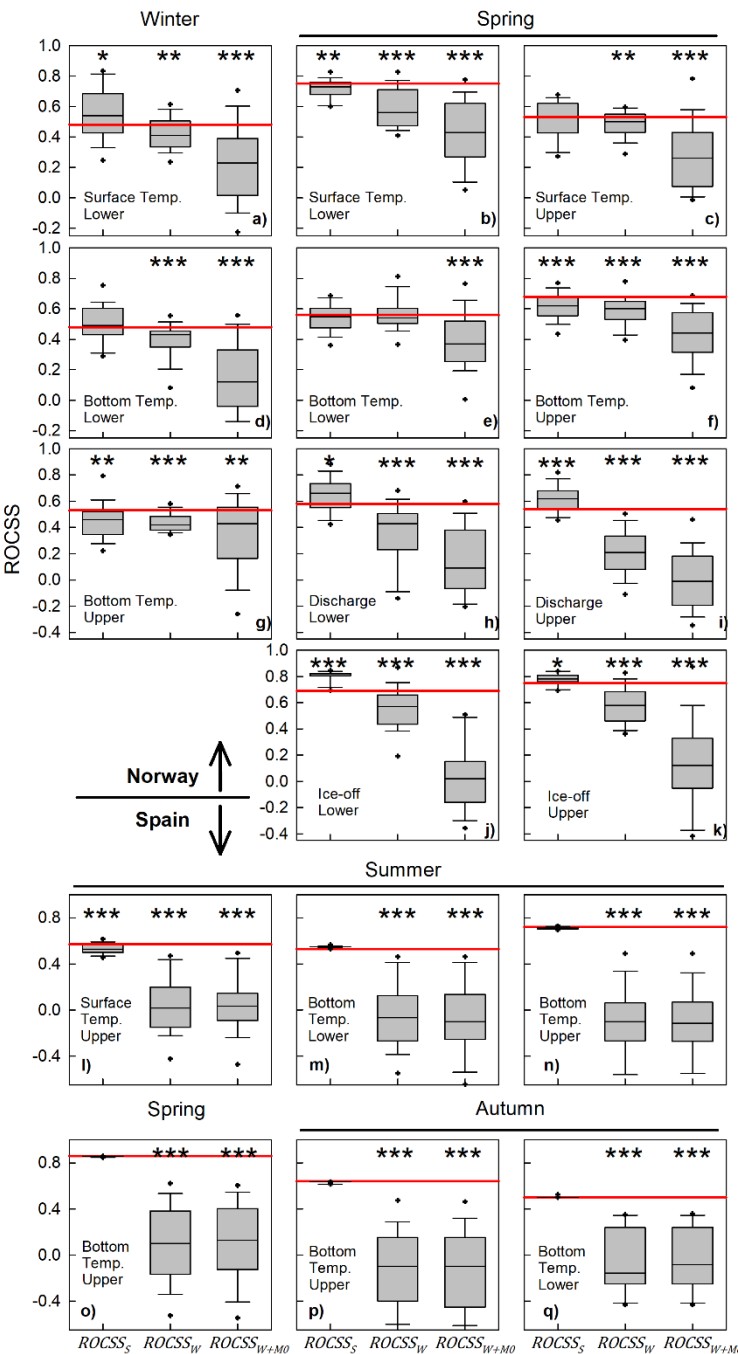

**Figure 4:** Box plots (n = 25) of $ROCSS_S$, $ROCSS_W$ and $ROCSS_{W+M0}$ from sensitivity analysis runs S-SA (replacing target season SEAS5 data with random data), W-SA (replacing warm-up ERA5 data with random data) and W+M0-SA (replacing warm-up period – ERA5 and first lead month – SEAS5 data with random data) for each window of opportunity at the Norwegian (a–k) and

Spanish (l–q) sites. $ROCSS_{original}$ is given by the red line, so $ROCSS_S$, $ROCSS_W$ and $ROCSS_{W+M0}$ below the red line indicate a loss of skill and values above the line indicate higher skill than the original forecast. ***, ** and * indicate significant difference between a given group of $ROCSS_S$, $ROCSS_W$ and $ROCSS_{W+M0}$ values and $ROCSS_{original}$ following Mann−Whitney Rank Sum test at a significance level of 0.001, 0.01 and 0.05, respectively. Note that the S-SA, W-SA and W+M0-SA were only performed for Sau Reservoir in Spain and Lake Vansjø in Norway because of the significant resources needed to perform this hindcast experiments.

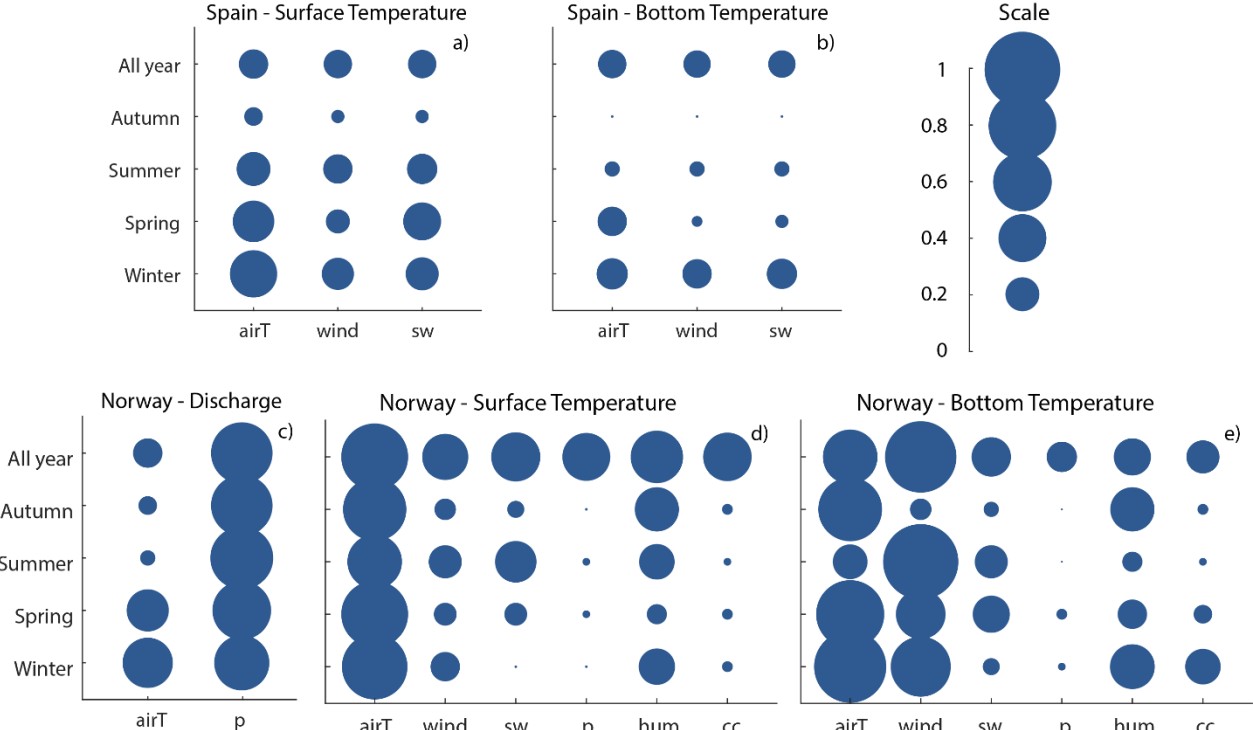

**Figure 5:** Relative sensitivity expressed as $1 − R^2$ of Lake_PO seasonal means to specific input variables estimated following the OAT-SA (see section 2.2.3 and Table 3 in the Methods section for details). Circle size represent relative sensitivity on a scale from 0 to 1, e.g., larger circle sizes, i.e., higher $(1 − R^2)$ values, indicate more influence of input variables on Lake_PO. Meteorological variable abbreviations: airT, p, wind, cc, hum, and sw stand for surface air temperature, precipitation, wind speed, cloud cover, relative humidity (or dew-point temperature) and short-wave radiation, respectively. Note that the relatively larger sensitivity of Lake_PO to specific input variables over the whole year can be larger compared to over a given season because of the strong seasonal cyclicity. Note that the OAT-SA was only performed for Sau Reservoir in Spain and Lake Vansjø in Norway.

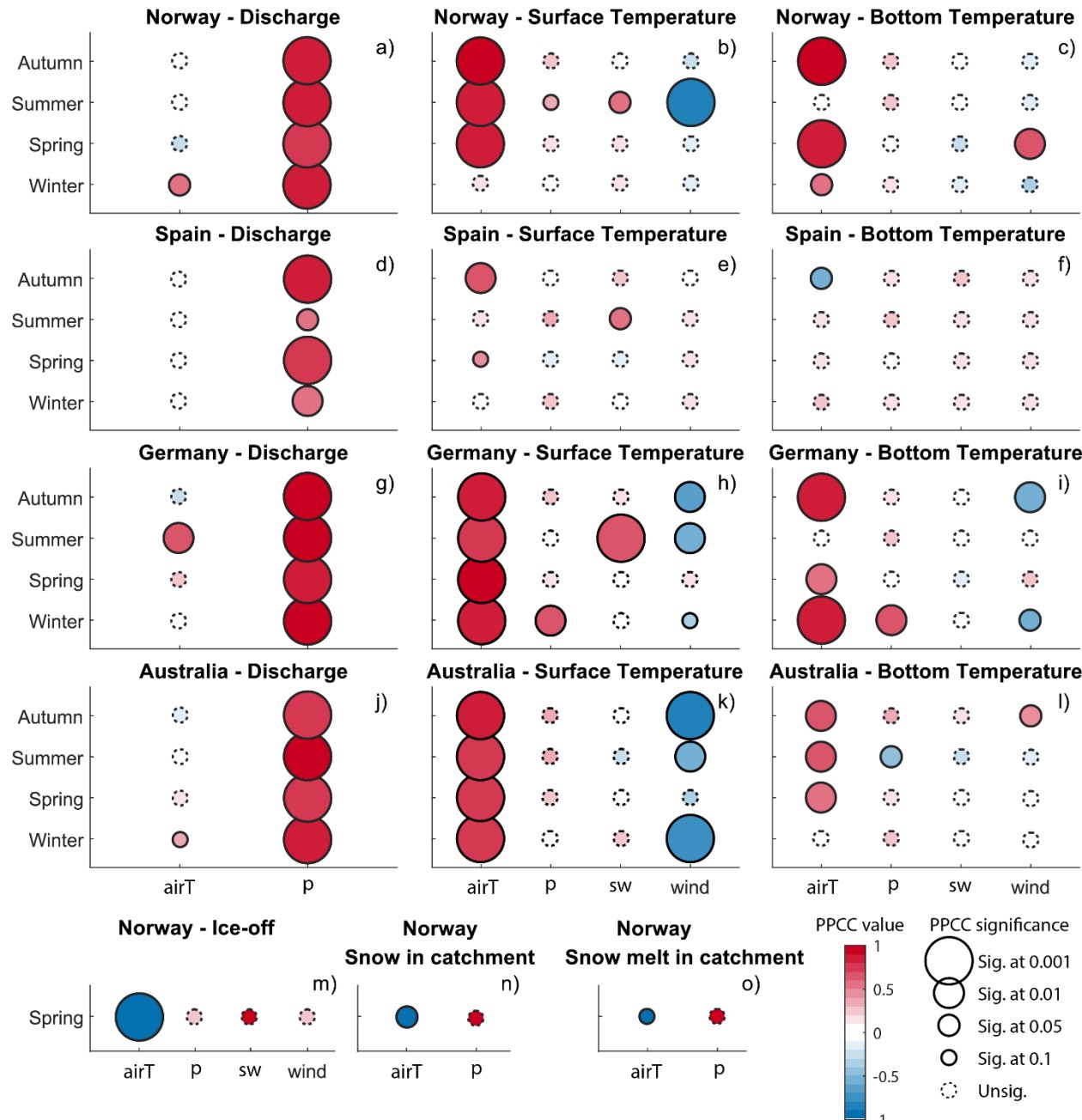

Figure 6: Pearson partial correlation coefficients (PPCC) between Lake_PO seasonal means and seasonal means of selected input variables. Circle color and size represent PPCC value (from -1 to 1) and significance, respectively. Meteorological variable abbreviations: airT, P, wind, cc, hum, and sw stand for surface air temperature, precipitation, wind speed, cloud cover, relative humidity (or dew-point temperature) and short-wave radiation, respectively. Only significance level at 0.1 or below were considered in the interpretation.

Next, we use SA outputs to better describe the origin of the prediction skill, considering inertia, time integration as well as variable interactions. Assuming that climate signals in the ERA5 and SEAS5 input data over the warm-up, first lead month and target periods are additive sources of prediction skill, we can use Eqs 1–3 to partition the prediction skill originating from those time periods, i.e., $P_{warm-up}$, $P_{M0}$ and $P_{season}$, respectively. For Sau reservoir, this calculation yields $P_{warm-up}$ of 0.94 to 1.0 leaving only an unsignificant fraction of prediction skill to the forcing data over the target season and the first lead month, as illustrated in Fig. 4. At this site, the output variables show in parallel very low sensitivity to input variables (Fig. 5 and 7) which supports a strong role of inertia or long-term time integration in hindcast predictive skill. The fact that 5 out of the 6 windows of opportunity are for bottom water is also consistent with inertia as the main source of skill given the low circulation rate and inertia of hypolimnions. For Lake Vansjø, Eqs 1–3 yielded $P_{season}$ of 0.003 (range: -0.19 to 0.18), $P_{M0}$ of 0.19 (0.04 to 0.37) and $P_{warm-up}$ of 0.29 (0.09 to 0.60). Hence, a significant fraction of prediction skill is originating from the SEAS5 boundary conditions although the largest source remains initial conditions through ERA5 data over the warm-up. Interestingly, the SEAS5 data over the first lead month is also a significant source of prediction skill. In fact, in decreasing order of importance, prediction skill originates from the warm-up, the first lead month and the target season. This progressive decrease in prediction skill is only observed at Lake Vansjø and suggests that across-variable integration of climate signals persists through the first lead month and, in some cases, the target season, but is progressively deteriorating as we move into the target season. Indeed, there is additional consistency between the SEAS5 input variables showing some forecasting skill and the output variables. In fact, surface, and bottom temperature in spring at Vansjø are sensitive to surface air temperature and wind (Fig. 6b and c), and surface aur temperature, wind u and v components are associated with some windows of opportunity in spring (Table 6). Similarly, ice-off is sensitive to surface air temperature, as are snow quantities and melt intensities in the catchment (Fig. 6m–o). Hence, in contrast to Sau reservoir where most of the prediction skill seems to originate from inertia, at Lake Vansjø, across-variable integration contributes to predictive skills.

## 4    Discussion

### 4.1  Sources of skill

Our investigation into relationships between input and output variables and the sensitivity of predictive skill to meteorological data inputs over different time periods have yielded important insights into the sources of seasonal lake forecasting skill in our case study sites.

A key finding is that predictive skill is mostly sensitive to meteorological inputs over the warm-up and first lead months (Fig. 4, Section 3.4), although some specific windows of opportunity are also somewhat sensitive to the meteorological data over the target season. Hence, integration of the climate signal over time or across variables by catchment hydrologic and physical processes, e.g., snow accumulation (Harrigan et al., 2018) or heat accumulation in lakes, is likely a key source of predictive skill. In fact, Mercado-Bettin et al. (2021) already noted an increase in prediction skill when moving from weather to discharge

to lake temperature, i.e., in an increasing order of time and across variable integration of climate signals. Strong inertia is also a potential source of prediction skill.

After accounting for forecasting skill from the forcing data over various periods (Section 3.3), a large proportion of the skill still remains unexplained, especially for some selected windows of opportunity at Lake Vansjø in Norway. Bottom water temperature at Lake Vansjø in spring shows the highest residual skill after removal of skill from warm-up and first lead month (Fig. 4e–f). Surface and bottom temperature show a different degree of coupling with air temperature. In fact, while surface temperature responds tightly to changes in air temperature (Butcher et al., 2015; Schmid et al., 2014), bottom temperature responds to a variety of complex interactions influenced by lake characteristics (e.g., fetch, surface area, depth, light penetration; Butcher et al., 2015). Indeed, bottom temperature in spring depends on preceding winter conditions but also on the intensity and length of the spring mixing event. To fully capture the intensity of this event, the model requires good initial water temperature inherited from previous winter but also skillful weather forcings, especially for surface air temperature and wind (Fig. 6c). In fact, for bottom temperature in spring to be higher than normal, it requires surface water to be heated up more than normal, mainly through heat exchange with air temperature, but also the lake to remain mixed for a longer time period than normal. The interaction between skill from legacy and from weather forcing might thus be another source of predictive skill. The fact that the proportion of forecasting skill progressively decreases from warm-up, through the first lead month and the target season at Vansjø suggests that the interactions between input variables, which are incorporated in the process representation within the models, provide some skill but progressively deteriorates as we move forward in time. At Sau reservoir in Spain, on the other hand, all skill is lost at the sharp boundary between the warm-up and the first lead month. This difference might be related to the presence of skill from the SEAS5 data at Vansjø (Table 6) and not at Sau. In other words, in the absence of skill in SEAS5 data, no additional skill can originate from interaction effects.

Literature on streamflow hindcasts broadly shows that beyond the first lead month, hindcasts forced with an ensemble of boundary conditions resampled from historical meteorology are typically more skillful than hindcasts driven by seasonal meteorological predictions (Arnal et al., 2018; Bazile et al., 2017; Greuell et al., 2019). Hence, better lake forecasting skills could likely be achieved by simply forcing our models with climatology. Our results partly fit with these findings, as the skill of S-SA hindcasts for selected windows of opportunity were higher than the original hindcasts (Fig. 4a, h–k). These S-SA hindcasts are similar to climatology-driven hindcasts, although they are associated with higher uncertainty since they are driven by random SEAS5 data and should therefore be regarded as a minimum forecasting potential. For some windows of opportunity, however, SEAS5 was a significant source of predictive skill (Fig. 4b, f, g and l). In those cases, only an improvement in SEAS5 forecasting skill is likely to improve lake forecasts. Improvement for only selected variables in SEAS5 would likely be enough to yield a significant increase in lake forecasting skill since most of the output variables presented here showed sensitivity to one or two input variables (Fig. 6).

### 4.2 Limitations and implications for seasonal lake forecasts

One apparent limitation of our study is the use of reanalysis weather data and pseudo-observations as inputs and benchmark output variables. Using pseudo-observations for skill assessment is a common methodology in streamflow forecasting studies (Alfieri et al., 2014; Wood et al., 2016) and it offers the opportunity to investigate the relationship between forecasting skills, initial and boundary conditions, while putting less emphasis on model errors and biases (Harrigan et al., 2018). Working with reanalysis weather data generates a less site-specific workflow and removes difficulties associated with dealing with temporal and spatial heterogeneity in observed data. Nevertheless, here we also evaluate the forecasting skill against catchment and lake observations when possible (Table 7) and show that most of the windows of opportunity reported for water temperature held while those for discharge are no longer significant compared to observations. This discrepancy between discharge and water temperature can be related to the fact that discharge tends to be more variable than water temperature, with short-lived high peaks which are difficult to model. The catchment models therefore performed less well than the lake models. This further suggests that evaluation against observations is likely more important for discharge than for water temperature.

The prediction skill of the seasonal lake forecasts can be influenced by multiple factors including the catchment and lake models used, the prediction skill of the forcing meteorological hindcasts, the quality and frequency of observations against which the models are calibrated, the nature of the system (e.g., potential for inertia), and the model calibration procedures. Given that we applied our workflow to only four case-study sites, unraveling the impact of all of the above-mentioned factors is out of the scope of this study and should be addressed through a more systematical application of our workflow to a larger number of sites. Our results rather highlight two opportunities for seasonal lake forecasting. First, prediction skill of the forcing meteorological SEAS5 hindcasts, expected to be stronger around the tropics, was the largest at the northernmost Norwegian site (Table 6) and effectively transferred from meteorological to lake hindcasts (Section 3.3). This highlights that, although the prediction skill of the meteorological forecasts is generally higher at the equator, there is not a monotonic decrease in skill with increasing latitude, rather there is high spatial variability in skill. Potentially useful seasonal meteorological and lake forecasts can therefore still be obtained at higher latitudes. Second, given that inertia and integration over time were the dominant sources of predictive skill at Sau reservoir and Lake Vansjø, useful hindcasts could already be issued without the use of SEAS5 data. In fact, our workflows show limited sensitivity to boundary conditions over the target season. Hence, future workflows should use selected climatology as forcing data over the target season, in addition to (or instead of) seasonal meteorological prediction. This benchmark forecasting workflow with climatology will likely yield similar or more skillful forecasts, as well as being less time-consuming to set up. Indeed, even with randomly selected years from the SEAS5 data, which can be seen as a highly uncertain climatology, some windows of opportunity are more skillful than with the correct SEAS5 data (Fig. 4). Nevertheless, if seasonal meteorological prediction products become more skillful, they will likely be a real asset for lake seasonal forecasting enabling additional skills through interactions over time.

State-of-the-art modeling practices typically involve calibrating hydrologic and lake models against daily observations. Nevertheless, daily observations of water quality are often not available or only cover a fraction of the time of interest. Table

7 illustrates the challenges related to data coverage and model evaluation where many calibration and validation statistics could not be estimated because of the lack of observations. In addition, calibrating to daily data prioritizes model parameterizations which are able to capture daily variability, but not necessarily seasonal or interannual variability, which are more relevant for seasonal forecasting. Calibrating the hydrologic and lake models using seasonal means or medians, in combination with daily data, could solve the observation coverage issue while improving seasonal predictive skill, but is then hampered by a low number of observed data points for calibration. Nevertheless, one needs to ensure that the seasonal averages are calculated from representative and well-distributed datasets. For Lake Vansjø, this would not have solved the lack of observations in Spring for example, because observations only cover April and May. For Sau and Wupper reservoirs, on the other hand, this would have been possible and potentially improve predictive skills. In any case, having access to more complete, long-term and systematic observations on water temperature, inflow and outflow discharge, including abstraction and over- flows for reservoirs, would facilitate robust model calibration and validation, and likely model predictive skills. The skill of water quality forecasting tools heavily depends on observation availability. Hence, continued efforts should be put on ensuring that observational programs are suited to providing the information needed by our models (Robson, 2014).

## 5  Conclusion

Lake seasonal forecasts could provide valuable knowledge for water managers to help protect drinking water reserves, as well as ecological and recreational services under increasing pressures from water demand, anthropogenic pollution, and climate change. Nevertheless, their use is still limited in the water sector. Here we unravel the source of predictive skill of lake seasonal hindcasts at four case-studies across Europe and in Australia, including inflow discharge, surface and bottom water temperature as well as ice-off dates. Through sensitivity analyses, we contribute to the demystification of lake forecasting tools with the long-term objective of facilitating their utilization in the water sector. In Spain, where the seasonal meteorological predictions have negligible skill, the source of predictive skill is mainly catchment and lake inertia. In Norway, where some seasonal meteorological predictions are skillful, predictive skill is coming from, in decreasing order of importance, inertia, time- and across-variable integration of climate signals through catchment processes, and seasonal meteorological predictions over the target season (SEAS5). In Norway, skillful SEAS5 meteorological hindcasts over specific seasons likely contribute to sustaining the predictive skill from antecedent conditions through to the target season.

Despite its central role in the probabilistic nature of the forecasting workflow, SEAS5 meteorological forcing data contributes little to the predictive skill, and often reduces the performance of the hindcasts. Hence, our findings suggest that using a probabilistic ensemble catchment-lake forecast without SEAS5 forcing data is currently likely to yield higher quality forecasts in most cases, as demonstrated by hindcasts driven with randomly selected SEAS5 data. Nevertheless, upon improvement in the skill of the seasonal meteorological forecasts, only a small step would be needed to provide more skillful lake forecasts for better water management.

**Index of abbreviations (in order of appearance)**

SEAS5: seasonal meteorological forecast dataset from the European Centre for Medium Range Weather Forecasts.

ERA5: meteorological reanalysis dataset from the European Centre for Medium Range Weather Forecasts.

NSE: Nash–Sutcliffe efficiency coefficient

KGE: Kling-Gupta efficiency coefficient

RMSE: Root-mean squared error

$R^2$: square of the Pearson correlation coefficient

Lake_PO: Lake pseudo-observations of water temperature, inflow discharge and ice-off produced with coupled catchment and

lake models forced with ERA5 meteorological data.

Lake_F: Seasonal lake hindcasts of water temperature, inflow discharge and ice-off produced with coupled catchment and lake

models forced with SEAS5 meteorological data (25 members).

M0: First lead month

M1–M3: Month 1 to month 3 of the lake forecast, i.e., target season of the lake forecasts.

ROCSS: Relative Operating Characteristic Skill Score

RPSS: Ranked Probability Skill Score

FRPSS: Fair (or unbiased) RPSS

$ROCSS_{original}$: ROCSS for Lake_F as compared to reference forecast based on climatology from Lake_PO.

$ROCSS_{Obs}$: ROCSS for Lake_F as compared to reference forecast based on local observations.

SA: Sensitivity analysis

S-SA: Sensitivity analysis of Lake_F to boundary conditions over the target season (M1–M3)

W-SA: Sensitivity analysis of Lake_F to boundary conditions over the warm-up period

W+M0-SA: Sensitivity analysis of Lake_F to boundary conditions over the period covering the warm-up and first lead month

$ROCSS_S$: ROCSS for Lake_F following S-SA as compared to reference forecast based on climatology from Lake_PO

$ROCSS_W$: ROCSS for Lake_F following W-SA as compared to reference forecast based on climatology from Lake_PO

$ROCSS_{W+M0}$: ROCSS for Lake_F following W+M0-SA as compared to reference forecast based on climatology from
Lake_PO

OAT-SA: One at a time sensitivity analysis

PPCC: Partial correlation coefficient

airT: Surface air temperature

airP: Surface air pressure

cc: Cloud cover

hum: Relative humidity (or dew-point temperature)

sw: short-wave radiation

lw: downwelling long-wave radiation

U: U-component of wind speed

V: V-component of wind speed

p: Precipitation

## Code and data availability

You can find all the code and data files related to this manuscript at: https://github.com/NIVANorge/seasonal_forecasting_watexr

## Acknowledgments

This study was largely funder under the WATExR project (https://nivanorge.github.io/seasonal_forecasting_watexr/), which is part of ERA4CS, an ERA-NET initiated by JPI Climate, and funded by MINECO-AEI (ES), FORMAS (SE), BMBF (DE), EPA (IE), RCN (NO), and IFD (DK), with co-funding by the European Union (Grant 690462). MINECO-AEI funded this research through projects PCIN-2017-062 and PCIN-2017-092. We thank all water quality and quantity data providers: Ens d'Abastament d'Aigua Ter-Llobregat (ATL, https://www.atl.cat/es), SA Water (https://www.sawater.com.au/), Wupperverband (www.wupperverband.de), NIVA (www.niva.no) and NVE (https://www.nve.no/english/). We acknowledge ECMWF for providing the SEAS5 and ERA5 data. We are grateful to two anonymous reviewers who contributed to significantly improve the manuscript.

## Competing interests

The contact author has declared that neither him nor their co-authors have any competing interests.

## Author contributions

RM, LJ-B, EdE, EJ, KR, LdvL, M-DF, and SH designed the study and provided guidance on modelling and forecasting approaches. FC, LJ-B, MNO, JS, DM, MS, AF, and TM contributed to the modelling, forcing data pre-processing and forecasting. FC, DM, MS, AF performed the sensitivity analyses. FC drafted the manuscript. All authors edited the manuscript.

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

Supplementary Material. [Data set]. Zenodo. https://doi.org/10.5281/zenodo.5906258

Jackson-Blake, L. A., Sample, J. E., Wade, A. J., Helliwell, R. C., & Skeffington, R. A. (2017). Are our dynamic water quality models too complex? A comparison of a new parsimonious phosphorus model, SimplyP, and INCA-P. *Water Resources Research*, *53*(7), 5382–5399. https://doi.org/10.1002/2016WR020132

Jeppesen, E., Pierson, D., & Jennings, E. (2021). Effect of Extreme Climate Events on Lake Ecosystems. *Water*, *13*(3), 282.

https://doi.org/10.3390/w13030282

Johnson, S. J., Stockdale, T. N., Ferranti, L., Balmaseda, M. A., Molteni, F., Magnusson, L., Tietsche, S., Decremer, D., Weisheimer, A., Balsamo, G., Keeley, S. P. E., Mogensen, K., Zuo, H., & Monge-Sanz, B. M. (2019). SEAS5: The new

ECMWF seasonal forecast system. *Geoscientific Model Development*, *12*(3), 1087–1117. https://doi.org/10.5194/gmd-12-1087-2019

Jolliffe, I. T., & Stephenson, D. B. (2012). *Forecast Verification: A Practitioner's Guide in Atmospheric Science*. John Wiley & Sons.

Labrousse, C., Ludwig, W., Pinel, S., Sadaoui, M., & Lacquement, G. (2020). Unravelling Climate and Anthropogenic Forcings on the Evolution of Surface Water Resources in Southern France. *Water*, *12*(12), 3581. https://doi.org/10.3390/w12123581

Lledó, Ll., Torralba, V., Soret, A., Ramon, J., & Doblas-Reyes, F. J. (2019). Seasonal forecasts of wind power generation. *Renewable Energy*, *143*, 91–100. https://doi.org/10.1016/j.renene.2019.04.135

Lopez, A., & Haines, S. (2017). Exploring the Usability of Probabilistic Weather Forecasts for Water Resources Decision-Making in the United Kingdom. *Weather, Climate, and Society*, *9*(4), 701–715. https://doi.org/10.1175/WCAS-D-16-0072.1

Manzanas, R., Frías, M. D., Cofiño, A. S., & Gutiérrez, J. M. (2014). Validation of 40 year multimodel seasonal precipitation

forecasts: The role of ENSO on the global skill. *Journal of Geophysical Research: Atmospheres*, *119*(4), 1708–1719. https://doi.org/10.1002/2013JD020680

Marcé, R., Rodríguez-Arias, M. À., García, J. C., & Armengol, J. (2010). El Niño Southern Oscillation and climate trends impact reservoir water quality. *Global Change Biology*, 16(10), 2857–2865. https://doi.org/10.1111/j.1365-2486.2010.02163.x

Marino, S., Hogue, I. B., Ray, C. J., & Kirschner, D. E. (2008). A Methodology For Performing Global Uncertainty And Sensitivity Analysis In Systems Biology. *Journal of Theoretical Biology*, *254*(1), 178–196. https://doi.org/10.1016/j.jtbi.2008.04.011

Mariotti, A., Baggett, C., Barnes, E. A., Becker, E., Butler, A., Collins, D. C., Dirmeyer, P. A., Ferranti, L., Johnson, N. C., Jones, J., Kirtman, B. P., Lang, A. L., Molod, A., Newman, M., Robertson, A. W., Schubert, S., Waliser, D. E., & Albers, J.

(2020). Windows of Opportunity for Skillful Forecasts Subseasonal to Seasonal and Beyond. *Bulletin of the American Meteorological Society*, *101*(5), E608–E625. https://doi.org/10.1175/BAMS-D-18-0326.1

Mercado-Bettin, D., Clayer, F., Shikhani, M., Moore, T. N., Frias, M. D., Jackson-Blake, L., Sample, J., Iturbide, M., Herrera, S., French, A. S., Norling, M. D., Rinke, K., & Marce, R. (2021). Forecasting water temperature in lakes and reservoirs using seasonal climate prediction. *Water Research*, *201*, 117286. https://doi.org/10.1016/j.watres.2021.117286

Müller, W. A., Appenzeller, C., Doblas-Reyes, F. J., & Liniger, M. A. (2005). A Debiased Ranked Probability Skill Score to Evaluate Probabilistic Ensemble Forecasts with Small Ensemble Sizes. *Journal of Climate*, *18*(10), 1513–1523. https://doi.org/10.1175/JCLI3361.1

Pagano, T. C., Wood, A. W., Ramos, M.-H., Cloke, H. L., Pappenberger, F., Clark, M. P., Cranston, M., Kavetski, D., Mathevet, T., Sorooshian, S., & Verkade, J. S. (2014). Challenges of Operational River Forecasting. *Journal of*

*Hydrometeorology*, *15*(4), 1692–1707. https://doi.org/10.1175/JHM-D-13-0188.1

Pechlivanidis, I. G., Crochemore, L., Rosberg, J., & Bosshard, T. (2020). What Are the Key Drivers Controlling the Quality of Seasonal Streamflow Forecasts? *Water Resources Research*, *56*(6), e2019WR026987. https://doi.org/10.1029/2019WR026987

Pianosi, F., Beven, K., Freer, J., Hall, J. W., Rougier, J., Stephenson, D. B., & Wagener, T. (2016). Sensitivity analysis of environmental models: A systematic review with practical workflow. *Environmental Modelling & Software*, *79*, 214–232. https://doi.org/10.1016/j.envsoft.2016.02.008

Piccolroaz, S., Healey, N. C., Lenters, J. D., Schladow, S. G., Hook, S. J., Sahoo, G. B., & Toffolon, M. (2018). On the predictability of lake surface temperature using air temperature in a changing climate: A case study for Lake Tahoe (U.S.A.). *Limnology and Oceanography*, *63*(1), 243–261. https://doi.org/10.1002/lno.10626

Piccolroaz, S., Toffolon, M., & Majone, B. (2013). A simple lumped model to convert air temperature into surface water temperature in lakes. *Hydrology and Earth System Sciences*, *17*(8), 3323–3338. https://doi.org/10.5194/hess-17-3323-2013

Portele, T. C., Lorenz, C., Dibrani, B., Laux, P., Bliefernicht, J., & Kunstmann, H. (2021). Seasonal forecasts offer economic benefit for hydrological decision making in semi-arid regions. *Scientific Reports*, *11*(1), 10581. https://doi.org/10.1038/s41598-021-89564-y

Robson, B. J. (2014). State of the art in modelling of phosphorus in aquatic systems: Review, criticisms and commentary. *Environmental Modelling & Software*, *61*, 339–359. https://doi.org/10.1016/j.envsoft.2014.01.012

Schmid, M., & Read, J. (2022). Heat Budget of Lakes. In T. Mehner & K. Tockner (Eds.), *Encyclopedia of Inland Waters* (Second Edition) (pp. 467–473). Elsevier. https://doi.org/10.1016/B978-0-12-819166-8.00011-6

Schmid, M., Hunziker, S., & Wüest, A. (2014). Lake surface temperatures in a changing climate: A global sensitivity analysis. *Climatic Change*, *124*(1), 301–315. https://doi.org/10.1007/s10584-014-1087-2

Soares, M. B., Daly, M., & Dessai, S. (2018). Assessing the value of seasonal climate forecasts for decision-making. *WIREs Climate Change*, *9*(4), e523. https://doi.org/10.1002/wcc.523

Staudinger, M., & Seibert, J. (2014). Predictability of low flow – An assessment with simulation experiments. *Journal of Hydrology*, *519*, 1383–1393. https://doi.org/10.1016/j.jhydrol.2014.08.061

Stefan, H. G., & Preud'homme, E. B. (1993). Stream Temperature Estimation from Air Temperature1. *JAWRA Journal of the American Water Resources Association,* 29(1), 27–45. https://doi.org/10.1111/j.1752-1688.1993.tb01502.x

Themeßl, M., Gobiet, A., & Leuprecht, A. (2011). Empirical-statistical downscaling and error correction of daily precipitation from regional climate models. *International Journal of Climatology*, 31(10), 1530–1544. https://doi.org/10.1002/joc.2168

Toffolon, M., Piccolroaz, S., Majone, B., Soja, A.-M., Peeters, F., Schmid, M., & Wüest, A. (2014). Prediction of surface temperature in lakes with different morphology using air temperature. *Limnology and Oceanography*, *59*(6), 2185–2202. https://doi.org/10.4319/lo.2014.59.6.2185

Troccoli, A. (2010). Seasonal climate forecasting. *Meteorological Applications*, *17*(3), 251–268. https://doi.org/10.1002/met.184

Troin, M., Arsenault, R., Wood, A. W., Brissette, F., & Martel, J.-L. (2021). Generating Ensemble Streamflow Forecasts: A Review of Methods and Approaches Over the Past 40 Years. *Water Resources Research*, *57*(7), e2020WR028392. https://doi.org/10.1029/2020WR028392

Werner, M., Cranston, M., Harrison, T., Whitfield, D., & Schellekens, J. (2009). Recent developments in operational flood forecasting in England, Wales and Scotland. *Meteorological Applications*, *16*(1), 13–22. https://doi.org/10.1002/met.124

Wilcke, R. A. I., Mendlik, T., & Gobiet, A. (2013). Multi-variable error correction of regional climate models. *Climatic Change*, *120*(4), 871–887. https://doi.org/10.1007/s10584-013-0845-x

Wood, A. W., Hopson, T., Newman, A., Brekke, L., Arnold, J., & Clark, M. (2016). Quantifying Streamflow Forecast Skill Elasticity to Initial Condition and Climate Prediction Skill. *Journal of Hydrometeorology*, *17*(2), 651–668. https://doi.org/10.1175/JHM-D-14-0213.1

Wuijts, S., Claessens, J., Farrow, L., Doody, D. G., Klages, S., Christophoridis, C., Cvejić, R., Glavan, M., Nesheim, I., Platjouw, F., Wright, I., Rowbottom, J., Graversgaard, M., van den Brink, C., Leitão, I., Ferreira, A., & Boekhold, S. (2021). Protection of drinking water resources from agricultural pressures: Effectiveness of EU regulations in the context of local realities. *Journal of Environmental Management*, *287*, 112270. https://doi.org/10.1016/j.jenvman.2021.112270

Yi, S., Sun, W., Feng, W., & Chen, J. (2016). Anthropogenic and climate-driven water depletion in Asia. *Geophysical Research Letters*, *43*(17), 9061–9069. https://doi.org/10.1002/2016GL069985

Zhu, S., Piotrowski, A. P., Ptak, M., Napiorkowski, J. J., Dai, J., & Ji, Q. (2021). How does the calibration method impact the performance of the air2water model for the forecasting of lake surface water temperatures? *Journal of Hydrology*, *597*, 126219. https://doi.org/10.1016/j.jhydrol.2021.126219

Zhu, S., Ptak, M., Yaseen, Z. M., Dai, J., & Sivakumar, B. (2020). Forecasting surface water temperature in lakes: A comparison of approaches. *Journal of Hydrology*, *585*, 124809. https://doi.org/10.1016/j.jhydrol.2020.124809