# Peer review of "Sources of skill in lake temperature, discharge and ice-off seasonal forecasting tools"

_Hydrology and Earth System Sciences, 2022_

## Author Comment (AC4)

Thank you for this thorough and very complete review. I apologize for the lack of precision in our terminology and any impact it might have had on your understanding of this study. We still hope there is some time for discussion if there is additional interest.

Yes, it is true that Hydrometeorological prediction was not our main research interest at the start of our project, although some of our co-authors and project partners are highly experienced scientists in weather forecasts.

We will make sure to use the standard terminology used in this field. Thank you for picking those multiple loose threads that needed further attention. We will make sure the terminology and writing style is unified throughout the manuscript.

Below are some preliminary responses to specific points the reviewer raised ("excerpts of the reviewer's comments are underlined for clarity").

windows of opportunity: these windows of opportunity are indeed not properly defined in the text. Mentioned in the introduction l. 89-93. A window of opportunity will be defined as a skillful forecast for a given variable's tercile during a given season. In other words, a trustful forecast that a stakeholder can use as a decision support information. For example, the lower tercile of surface temperature in Spring at our Norwegian site is a window of opportunity (Table 4). In other words, our workflow is able to forecast when surface temperature will be lower than normal in Spring at the Norwegian site. We focus on those because the whole project and study here is to be seen from a stakeholder point of view, and we would like to focus only on forecasts that we are confident are the most trustful. This point will be made clearer from the start, to also respond to your comment on l. 194-197.

Hindcasts: Sorry for the confusion here. Yes we used hindcasts for the 92 three-month seasons (11/1993 to 11/2016) (see L. 170). We refer to 1994-2016 for simplicity, but we will make sure that everything is clear and consistent throughout the manuscript, also when the hindcasts were produced.

Another point is the source of predictability. We agree that the SEAS5 forecasting system cannot be seen as a source of predictability, it likely further transfers predictability from e.g., ENSO and NAO, as you mentioned, to our catchment and lake model. We will consider this point very carefully as it impacts the title as well. Thank you for picking this up. We will make sure to avoid these confusions throughout the manuscript (e.g., L. 53-55; L. 76-77 …) and any other confusion caused by unprecise terminology related to forecasting (e.g., l. 167-168; l. 173-174; L. 174-175; l. 191; l. 347-348; l. 366-368). We will avoid any cryptic formulation.

SEAS5 system and ERA5 resolution and downscaling issues: We have used standard tools and have colleagues in the group that are experienced with this type of issues, so I know we have bias-corrected, down-scaled ERA5 and SEAS5 data in an acceptable way. We will provide further detail on these pre-processing steps in the revised version of the manuscript. For now, you can have a look at Mercado-Bettin et al. (2021; https://doi.org/10.1016/j.watres.2021.117286)

I'm really grateful to the reviewer for the quality and completeness of this review, Thank you for these specific and relevant comments. I'm trying to provide preliminary (for now) answers to some of the comments below:

More specific comments

Introduction

Line 24-26: Do you mean the skill of the meteorological predictions (SEAS5 outputs) are worse then the skill of the hydrological predictions?

       Yes this is what we mean here. Meteorological predictions were worse than the skill of the discharge (to only some extent) and lake water temperature predictions.

Line 70 "...temperature predictions and forecasts" and L. 72

Yes thank you, we will make sure to have a consistent terminology

Line 80-84: I do not understand what the authors try to argue, what is meant by "water flow predictability", do you mean discharge of the rivers? Lake level heights? Can you make this sentence clearer?

Yes we meant river discharge. We will make that clearer.

Line 90. We will introduce ice-off.

Line 90-91 and Table 4 (showing actual values of ROCSS and FRPSS) and Table 5

The aim of this table (Table 4) was rather to quickly show where and when the forecasts were skillful, and compare the skills of SEAS5 forecasts (climate predictions) with lake forecasts. We can provide all values in the supplementary to avoid overloading the table. Note that we haven't considered all forecasts with ROCSS and FRPSS higher than 0 as "skillful". We have described this L. 194-197:
"Threshold RPSS and ROCSS values above which RPSS and ROCSS are significant at 95% confidence are calculated by built-in VisualizeR functions and were used to identify windows of opportunity (i.e., combinations of seasons, variables and terciles for which forecast performance was significantly better than the reference). In our case, these thresholds typically range between 0.47 and 0.55."

Admittedly, our formulation is lacking clarity. We will provide more details in the revised MS in the text and the table caption. This is also linked to your comment on L. 262 on how fair we determine a RPSS is significant and Table 5.

Methods:

Line 101 to 111: Thank you for your nice suggestions. We will incorporate those.

Line 114-115: As stated above, we will provide more detail on the bias-correction method used (also from L. 125).

Line 116: term "impact models" and "impact variables".

Yes these are the water quality models and variables. We will unify these terms and them throughout the manuscript.

Line: 125

Unfortunately, we did not do any forecast verification with different bias-correction. Given the frame of the project and the resources it takes, it will unfortunately not be possible. On the other hand, we can look at additional scores and measures.

Line 127-128:

We apologize for the messy abbreviations of variables here, we will make sure those are defined and consistent throughout the manuscript. Given that we applied different models at the four sites, sometimes the forcing weather variable were slightly different. We will make sure to described that and harmonize everything.

Line 130-131: Yes we can provide further detail on the observations, these were collected by our institutes, or published datasets.

Line 143-145:

All hydrological models were calibrate against local observations, this is described in Mercado-Bettin et al. (2021; https://doi.org/10.1016/j.watres.2021.117286) but will be briefly described here again for clarity. Note that models were calibrated and validated against two different time periods following best-practices.

Line 152: Thank you, we will do.

158-159: "Most common statistical goodness-of-fit parameters, e.g., Kling-Gupta efficiency (KGE), NSE and RMSE, for hydrological and lake modeling were calculated."

We will explicitly describe the values of these performance criteria.

Line 174-175: Aggregation prior to forecast verification

We don't do any lead-time dependent verification, we only aggregated the forecast to seasonal means (1 value for the whole 3-month period for each year between 1993 and 2016) and used that in our verifications and calculations of skill scores. We will make sure to clarify this point in the manuscript.

Table2 and full paragraph:

Thank you for your suggestions here. Yes pseudo-observations were used as the reference forecast. In addition, we used also observations as the reference forecast, when and where it was possible (when data gaps where below a given threshold). We apologize again, this threshold was not described properly. Note however, that the "Obs coverage" in table 5 highlight the cases for which there are enough observations. This point will clarified.

Line 200-227: I struggle a lot to follow the explanation. First of all what is ROCSS**s/** ROCSS**w/** ROCSS**w+t**

Thank you for your suggestions on rephrasing the paragraph, confused formulations and updating the paragraphs' titles. These ROCSS are the skill scores calculated from the forecasts of the various sensitivity analyses (SA) described just before, where forcing data over the target season (S), the warm-up period (W) and warm-up plus transition periods (W+T) was replaced by random data. As described l. 216-217: "The outputs of S-SA, W-SA, and W+T-SA were used to produce tercile plots and calculate ROCSS. The comparison of the ROCSS values ($ROCSS_i$) obtained for the various SAs" We will clarify this formulation to make sure this point is not confused.

Line 240-244:

This will be clarified. In fact, we will provide lake heat budgets for each site to support this, as it was also raised by the other reviewer. This is supported by our data analysis and also shown in the literature (e.g., Blottiere, 2015).

Results

Line 251: We will add part of Table S2 into the main text.

Line 262: Sure, we will provide more background on fair RPSS and what it accounts for.

Line 269-270:

"Only 0 to 10% of the SEAS5 climate 270 hindcasts are skillful, on average".

I struggle understanding this result. The RPSS (or any score) is usually determine based on a large sample forecasts (or hindcasts). Of course, for an individual forecast this might be poor but the full picture of forecast performance can only be revealed when the Scores for many issued forecasts are investigated.

Here RPSS and ROCSS are determined based on the 23 seasonal means (spring, summer, autumn, winter) from 1993 to 2016. As described l. 189-191, Briefly, the RPSS provides a relative performance measure on how well the probabilistic ensemble is distributed over the lower, middle and upper terciles, while the ROCSS provides a relative measure of discriminative skill for each category.

I think here you should add a figure with the results where the reader can see how the hindcast performance actually is. From the text and the table alone, it is rather difficult to follow your argumentation. In addition, what is the definition of a skillful hindcast in your context? Is every hindcast with a ROCSS>0.5 skillful or do you use other thresholds? It would be crucial to mention the numbers at least once in the results section as well.

We were struggling to find a clear and concise way to create a figure showing hindcast performance but we were looking for this. So we would be very happy if you have a suggestion? Maybe a figure from a published paper than we can be inspired from. How would you like to see this information plotted?

Regarding skillful forecasts, we will clarify this. We basically consider that all hindcast with significant ROCSS are skillful forecasts (where the threshold is in between 0.47 and 0.55 (as described above).

Line 278-280: How many seasons are discarded due to missing observations? Please indicate the exact numbers such that the reader knows how many samples (hindcasts) are actually used for the analysis. Or at least refere to the table where the numbers are listed.

We calculated ROCSS_obs only when >50% of the seasons (i.e., 12 seasons) were represented by some data. But in practice, most of the variables for which we calculated ROCSS_obs had 100% (i.e., 23 seasons) covered by observations. See also "Obs coverage" in Table 5. We will introduced this notion of observation coverage earlier in the manuscript and make sure everything is clear how we calculated ROCSS_obs, including the number of seasons.

Line 283-286: Yes we will rephrase here for clarity.

Table 4

Again general comments: abbreviations are non intuitive. FRPSS is not introduced before. I do not get the message of this table. I would prefer to see the skill scores (e.g. as boxplots) over different seasons for the variables. It is not clear to me what temporal aggregation is the baseline of this analysis.

The main message here is to show that there is limited skill of the SEAS5 forecasts (climate predictions) but still some skill in the lake forecasts. In many cases, those are not synchronous, e.g., in Summer in Spain, 5 variables' tercile are associated with significant ROCSS, what we call "windows of opportunity", whereas there are only 2 weather climate variables' terciles with significant ROCSS. Looking at this table, we were hoping that the reader would say, "Oh, there is forecasting skill coming from somewhere else that SEAS5, e.g., inertia"

Regarding temporal aggregation, this is again based on seasonal means. We apologize for the oversight, and will include that in the caption.

Line 305:

Fig. 31 should be Fig. 3 This is Fig. 3l with "l" for "Luke Skywalker" for panel "l". We will use capitals to avoid misunderstanding.

Can you elaborate how the ROCSS is determined, what exact values are taken into account? Do you use daily values to calculate the scores or weekly/seasonal aggregated values? This is still unclear after reading the manuscript.

ROCSS are determined from seasonal means. We will make that point clear throughout the manuscript.

Fig 3: it is confusing that in the plot description and the text you mention ROCSSs etc. but on the x axis S-SA W-Sa etc are displayed. I suggest to unify all and make the plot more readable.

We will replace "S-SA", "W-SA" and "W+T-SA" with their respective ROCSS_i expressions.

Line 320-322:

Maybe because I do not understand what the windows of opportunity are, I do not get the message here. I suggest to more explicitly formulate what the impact is of changing the initial conditions and the forecast input. The result indicates that it is not worth using seasonal forecasts at all, which is hard to believe. Please elaborate such that the message gets clearer.

We will describe better what is the impact of changing the initial and boundary conditions. We suggest that there is a lot of skill that originates from legacy effects from the catchment model, and form inertia of the lake/reservoir systems themselves. Only at the norwegian site for specific variables and seasons, there are signs that using seasonal weather predictions provide further skill. For the other sites, we cannot see any sign of this. So, yes for most of the forecasts that were significantly skillful, it was not worth using seasonal forecasts to force the models. We will make this point clearer.

Line 323: again the title does not reveal to me what will be considered in this paragraph. I suggest to use a more appropriate title for this paragraph.

Thank you, we will provide a more relevant title.

Figure 4: It is hard to follow what is shown here. What is the relative sensitivity. Maybe it helps if you refer to the exact paragraph number in the label of the figure. In addition, is there a reason why you use a color coding and a size coding? I suggest either using color or size, otherwise it seems that multiple aspects are coded.

Yes, we will refer to the exact paragraph number for the description of how relative sensitivity is estimated, see L. 229-234, section 2.1.1. And you are right, color and size show the same information. We will keep only the size coding.

Line 347-348: "Hence, a significant fraction of predictability is originating from the SEAS5 dataset although the largest source remains ERA5 data over the warm-up". This sentence illustrates what I think makes the manuscript complicated to follow. The reader himself must make the connection what this means. If I am correct, it shows that the initial conditions are more important than the driving meteorological predictions. Is this correct? It would make the manuscript much more readable if you directly refer to formulations what it actually means in addition to just give such "cryptic" explanations.

Yes you understood this sentence right, and again we apologize for the cryptic formulations. Learning the forecasting terminology as we go. We will improve that.

Line 394-395:

Literature on streamflow hindcasts broadly shows that beyond the transition month, climatology-driven hindcasts are typically 395 more skillful than hindcasts driven by seasonal climate predictions (Arnal et al., 2018; Bazile et al., 2017; Greuell et al., 2019).

This is a misleading interpretation, when you say "climatology driven hindcasts". In all three publications a well established ESP (ensemble streamflow prediction) approach is used. This can be seen as a climatology driven hindcast, but for a scientific publication I would expect to have a clearer formulation. In addition, in these papers there is no transition month mentioned, a

concept I do not understand. All papers mention the first lead time month. What exactly is the transition month in your analysis?

We will be more precise when referring to these studies. The main point here is that hindcasts driven by seasonal climate predictions are not necessarily more skillful.

We will replace the "transition month" with lead month 0, in agreement with Greuell et al., 2019, i.e., the month following the date on which the forecast would have been issued. If the forecast was issued on February 1st, 2016, everything before that date is part of the warming period, February 2016 is what we used to call the "transition month", lead month 0, and the target season is the lead month 1 to 3: spring 2016, march to May 2016. This clarification will be included in the method section.

Again, thank you for this thorough review and for giving us the opportunity to improve by kindly pointing to our lack of precision in our terminology. Most of reviewers would just not bother to provide such constructive feedback.

---

## Author Response (AR1)

**Original title: Inertia and seasonal climate prediction as sources of skill in lake temperature, discharge and ice-off forecasting tools**

**Revised title: Sources of skill in lake temperature, discharge and ice-off seasonal forecasting tools**

Editor's and reviewers' comments are underlined for clarity. Line numbers refer to line in the revised (clean) version of the manuscript, if not specified otherwise.

**Editor:**

Both reviewers have indicated that this is an important topic, but they have also raised important issues requiring considerable work.

Thank you for responding to both reviewers. However, your responses remain very descriptive, and it is challenging to assess whether the changes will sufficiently address the issues raised by the reviewers. Together with the revised manuscript, please provide a detailed point-by-point responses to all reviewers' comments.

Thank you for your time and consideration. We provide below a point by point response to your comments and to each reviewer's comments.

I have finally highlighted two points that I believe need special attention when working on the revised manuscript:

* clarity. Both reviewers have stressed that the clarity, languages and terminologies have not reached a publication level. See specifically the detailed comments from RC2.

We have taken special care in rephrasing many imprecise sentences and have tried our best to use the most precise terminology. We believe the manuscript has been improved significantly.

* Retention time. Given the short retention time of the selected lakes, the advective heat might be highly important (see RC1). Is it realistic to not take into account the heat change from the throughflow? See for instance:
Schmid, M., & Read, J. (2022). Heat budget of lakes. In T. Mehner & K. Tockner (Eds.), Vol. 1. Encyclopedia of inland waters (pp. 467-473). https://doi.org/10.1016/B978-0-12-819166-8.00011-6
Råman Vinnå, L., Wüest, A., Zappa, M., Fink, G., & Bouffard, D. (2018). Tributaries affect the thermal response of lakes to climate change. Hydrology and Earth System Sciences, 22(1), 31-51. https://doi.org/10.5194/hess-22-31-2018

We had overlooked this in the previous version of the manuscript, but indeed, we have taken inflow water temperature into account in the heat bugdets. A side analysis, for review only, shows that the assumption the reviewer 1 raised doesn't hold. See our response to his comment for additional details.

**Reviewer 1:**

This article treats the important and so-far under developed field of seasonal forecasts (here 4 months into the future) of lake and drainage area properties including water temperature, ice-off, and river discharge. The authors combined two lake models (simulating surface and bottom temperature and ice cover, two lakes each) with four Hydrologic models (simulating discharge, one drainage area each). The method was applied at four lake-river system located in Norway, Spain, Australia, and Germany. Modeled systems include lakes spanning 19 o 60 meters depth and with a retention time from 0,2 to 1,1 years.

The coupled model setup was calibrated towards measurements (lake temperature and river discharge) and forced with reanalyzed data from ERA5, I would define this as a general circulation model (GCM). Hydrological-lake model performance was evaluates with KGE, NSE and RSM. Thereafter calibrated models was spun-up during one year and forecasted discharged and lake surface and bottom temperature during four months (one month initialization) spanning 13 years (1993-2016). Future forcing comes from 25 forecasts from the global forecasting tool SEAS5. SEAS5 was bias corrected and downscaled (grid adjustment) towards ERA5 to enable comparison.

The correctness of the forecasts (Lake_F) was evaluated trough a sensitivity analysis, comparison of Lake_F towards in-situ measurements and towards daily pseudo-observations (Lake_PO, daily output from the coupled hydrological-lake model setup forced with ERA5).  The end product of this manuscript consist in an evaluation (sensitivity analysis), of forecasting correctness for each river-lake system and a evaluation of forcing parameters influence on forecasts.

The manuscript show potential but is lacking in some areas which I list hereunder.

Thank you for this thorough and helpful review. We have carefully rephrased many imprecise sentences and have streamlined the manuscript. We also have added details on the case-study sites, observation data, our modelling procedure and how we dealt with inflow water temperature. Below are our responses for each of your specific comments**.**

**Chosen drainage areas and lakes**

The authors put forward that seasonal predictions work best next to the equator and worsen with increased latitude (line 50 to 58). Yet, no system was chosen in this region, Spain being the closest. The manuscript could still benefit from an analysis of latitudinal effects for the used forecasting method to improve forecasting towards the North/South pole.

Unfortunately, this is out of the scope of this study. We only have looked at four case-studies outside the tropics to investigate any opportunities and have "ready to go" workflow when seasonal meteorological predictions improve significantly. See L. 126-129

Additionally, the river-lake systems chosen contain lakes with very short retention time, i.e. big impact of rivers on water constituents, including temperature.  The model method used include the effect of changing lake volume, but not the effect of heat being transferred into the lakes by upstream drainage area (input temperature I could not find). Thereby it is reasonable to assume that the lake models (through calibration) had a better connection between surface and deep waters than is the in-situ case. Could this show up in your analysis of forcing parameter

importance ("Tracing of forecasting skill" section 3,4, Fig. 4)? This needs to be addressed/analyzed since you link forcing to lake processes, which in fact could be caused by upstream heat fluxes in the drainage area and not in the lakes themselves.

Admittedly, this was not clear in the manuscript. Nonetheless, we did take water temperature of the inflows into account. This is now announced early in the methods section, see L. 117-118 and described in detail at L. 207-216.

As a side-analysis, for review only, we have looked into the correlation coefficients between lake surface and bottom temperature for observations and pseudo-observations (modelled temperature with ERA5 data as forcing data). This analysis revealed that the correlation for pseudo-observations was not necessarily higher than for observations discrediting the reviewer's assumption that "lake models (through calibration) had a better connection between surface and deep waters than is the in-situ case" and giving further confidence that the heat lake budget is robust. In fact, the correlation coefficients for both observations and pseudo-observations ranged between 0.22 and 0.96 depending on the season and case-study. In given seasons, the observations even showed higher connections between surface and bottom water than pseudo-observations.

Now, we don't believe that adding inflow temperature in our analysis in section 3.4 would add any significant insight since inflow temperature is largely based on air temperature which is already accounted for. In order to avoid any redundancy in our analysis and following principles of parsimony. Note that we have added an assessment of the various lake heat fluxes in the supplementary information and refer to it at L. 330.

**Data**

This manuscript use ERA5 reanalysis as a stand in for in-situ measurements. Why is this, due to large spatial extent of drainage areas? If possible, show how this influence your modelling locally, or refer to documents where the reader can find this comparison between ERA5 and in-situ measurements, in best case for the regions being analyzed.

We forced our models with ERA5 meteorological data to ensure that our workflows were comparable between each case-study and future transferability of these workflows. We clarify this now at L. 126-129.

In addition, weather observations covering the whole range of variables needed to force our models were not available over the whole period from 1994 to 2016. Yes, this has likely influenced our modelling locally. Note however that, for ground truthing, we include a forecast verification step compared to a reference forecast based on observations which is not often included in forecasting studies (Table 7).

**Clarity**

The manuscript could benefit greatly from an index defining the many acronyms used, as well as improved description of tables and figures. Ex Table 4 and 5 is hard to understand.

Thank you. We have added an index at the end of the manuscript. L. 577.

Furthermore, I could not find/understand if the drainage area and lake models are coupled in time (run simultaneously), or if the drainage area models where run in advance to provide discharge for the lake models.

This is now clarified L. 114-118

**The language**

Certain words in the manuscript cause some confusion. Bellow I have stated some that might need to change

Skill – is associated with people. A fast car (a tool) has no skill it has performance, the driver on the over hand has skill. That said, I know skill is used more commonly to describe models (tools) in meteorology than hydrology. So I suggest that you define what you mean by skill if you want to keep this formulation.

Skill is now defined twice in the introduction (L. 53 and 88-89) and repeated in the results (L. 347)

Climate & climate prediction – studies involving effect of climate focus on longer time periods (>30 years) than what is the focus in this study (<1.5 years). Both SEAE5 and ERA5 comes from global GCM models, which could be used for climate studies. But in the context of this manuscript I do not think this is the right phrase describing the models you used.

Agreed, there was some confusion between climate and meteorological variables. This is now clarified throughout the manuscript. We rather refer to "seasonal meteorological forecasts/predictions" and mention explicitly the variables, when possible.

Hindcasts – is usually used in the setting of running models with data from past events, close to reanalyze with the aim to improve said models. Here this word is used in combination with SEAE5 forecast simulations. The authors have adjusted these to ERA5 (real data proxy) but the intention is still to use SEAE5 as forecasting forcing. Therefor consider other alternatives in the manuscript, or define this word in the context of your manuscript.

Hindcast is now properly defined in the introduction, see L. 93

Water quality – for drinking water and the biosphere, temperature is considered an important water quality parameter. Here we do not look at lakes and rivers in this sense, water quality one would assume here to entail dissolved constituents (nutrients, oxygen…). To avoid misunderstanding, consider using something else.

Agreed, we have replaced water quality by relevant alternatives throughout the manuscript, e.g., water temperature, lake.

Line 19 : "as previously  presented". Avoid need for reference in abstract.

We have removed this part from the sentence.

Line 67 : Consider adding the following reference https://doi.org/10.1016/j.watres.2020.115529

Thank you!

Line 72 to 74: partly untrue, air2water can run perfectly with seasonal forcing as you do here (only air temperature as forcing), and ice-off is currentl available indirectly.

The sentence has been updated. L. 71-72

Section 2,1,1 : The reader do not know where the lakes and drainage area (rivers) under investigation is situated. Add a map showing the global location and regional extent of each drainage area-lake system (rivers and lakes). Additionally add these system details (names, stations, etc.) where appropriate, ex. Table S1.

We have added a map (new Fig. 1) and detailed catchment maps are given by Jackson-Blake et al. 2022 as well as more background information on the case-studies. We now refer the readers to this study. See L. 110-111.

Line 111 or 112 : add reference: SEAS5: the new ECMWF seasonal forecast system. Stephanie J. Johnson, (2019), https://doi.org/10.5194/gmd-12-1087-2019

Done

Line122 to 123 : „Climate data where downloaded….". What do you mean in this sentence, ERA5 and/or SEAS5?

This sentence has been removed and we now provided much more details on ERA5 and SEAS5 data pre-processing steps, as requested by the other reviewer. See L. 139-150

Line 139 ")" missing

Thank you, Corrected.

Line 156 to 157 : Add details (equations and ex. RMSE) of this linear regression between in-versus outflow.

This is now described in more details L. 195-206 and in the supplementary material.

Figure 2. consider showing mean of SEAS5 predictions and ERA5 at the same time (i.e. continue black lines into transition and target season).

We already show the mean of SEAS5 predictions, adding a line would overload the figure (now Fig. 3). Besides, the main concept of this figure is to illustrate the input data used to force the model. ERA5 data was used only over the warm-up period.

Line 185 : RPSS looks to be missing from table 2 and table 3.

FRPSS was not considered for formal forecast verification because it doesn't allow to distinguish forecast performance for a given tercile. With these low performance lake forecasts that we have reported, this FRPSS didn't seem very useful to us. However, we still include it in Table 6 for reference.

Line 235 to 238 : something is missing here, hysteresis should make linear relationship between ex. air temperature and water temperature rather bad. Describe how good these linear fits were (in appendix). And/or show with figure and improve explanation.

Pearson partial correlation coefficients (PPCC) are calculated from seasonal means, and not daily values, which likely yielded much cleaner correlation than expected from daily values. This point is made clear now see L. 321.

Line 243 the reader are not familiar with the contributions of local heat fluxes at the chosen locations. Before disregarding for example cloud cover from the analysis, show the reader in numbers (or preferably figure as appendix) for each lake the seasonal heat budget contributions. I.e. uptake and emission of infrared longwave radiation, evaporation + condensation, sensible heat flux and uptake of surface downward solar radiation. Throughflow you only have the outflow (at some lakes?) since inflow temperature is missing.

This background information is now described L. 207-211 and we refer to the supplementary information L. 330 for further details.

Line 250 : RMSE not consistent with RMSE/sd in Table S2. What is RMSE/sd? Use the same in text as in Table S2.

This is now clarified, these first performance measures are for the whole year (now in Table 4) while Table S2 shows measures by season. RMSE/sd is now defined in the caption of Table S2.

Table 5. move description of asterisk under table and improve the site representation. Now you can not see what belongs to which system. And define the season duration.

Done.

Figure 3 and 4 : missing Germany and Australia, add or explain.

Note that we precise that the four probabilistic sensitivity analyses, S-SA, W-SA, W+M0-SA and OAT-SA were only performed at the sites in Spain and Norway because of the significant resources needed to execute these hindcast experiments (see L. 296-298). This point is now repeated in the figure captions (now Fig. 4 and 5).

Figure 4 : Something do not add up in your analysis. Top row for Spain – Bottom temperature, and Norway - Surface temperature appear to be to large compared to the individual season values taken together. I.e. if the impact is small most seasons, I do not see how it could e much larger on an annual basis.

We agree with you that it can appear quite surprising that the sensitivity calculated on an annual basis is much larger that over each single season. However, Norway's climate is subjected to strong seasonality which can be well captured over annual scales, but still have low correlation for a given season. To make this point clearer, we added a sentence in the figure caption (now Fig. 5). See L. 451-454.

Figure 5 : Why so many data gaps? Consider showing seasons where significance is worse (higher) but clearly state which significance level you trust.

This figure has been updated to show all correlation coefficients, even those that are not significant, and we now precise the significance level we trust in the figure caption (now Fig. 6) at L. 458-459.

Line 468 : add author contributions. Who did what?

Done, see L. 626-629

**Reviewer 2:**

The study assesses whether seasonal (meteorological) predictions can be used within a hydrometeorological forecasting tool to produce skillful lake temperature, discharge and ice-off forecasts. In general, the topic can be of interest for the community and gives a nice example of how seasonal (meteorological) predictions can be combined with hydrological models to produce application-relevant and user-specific outputs that provide a baseline for decision makers.

However, I struggle with the manuscript in its current form. Hydrometeorological prediction systems are per definition interdisciplinary as it combines meteorological and hydrological models. It is further complicated when the output is further used by decision makers. When reading the manuscript I got the impression that the authors do not have a strong background in hydro-meteorological forecasting systems and use terminologies that are very uncommon in this field. These imprecise formulations make the understanding of the manuscript very difficult. I suggest the authors to (further) familiarize with the literature body about hydrometeorological forecasting systems and to follow more closely the terminology used in standard literature.

In general, the language should be much more precise and terminology should be used from the community. Overall, it seems a bit like the manuscript is a composite of multiple different text/styles. I suggest unifying the manuscript to enhance its readability. Use commonly used terminologies throughout the manuscript (abbreviations, model descriptions). Delete obsolete/imprecise sentences or specifically mention the related numbers and where they can be found (e.g. 158-159: "…scores for hydrological and lake modlling were calculated." But there is no indication where in the manuscript the scores are provided)

We are very grateful to the reviewer for this thorough and very complete review. We have now carefully revised the manuscript with emphasis on using adapted terminology from literature, adding details in the methods and approach description focusing on clarity (including description of the bias correction as well as model and forecast verification). Please find below our responses to each specific comment.

Necessary clarifications:

windows of opportunity: From the manuscript I do not understand how "windows of opportunity" are defined. As this is a crucial baseline for the analysis, the authors should make an effort to properly describe how these windows are selected and what exactly the temporal resolution (or aggregation) of the forecast data is that they use to calculate the scores.

These windows of opportunity were indeed not properly defined in the text. We have now removed this term from the abstract because it adds confusion in such a short and concise text. However, we have introduced it early in the manuscript and carefully implement its description throughout the manuscript. See L. 99-104, 274-276 and 346.

Hindcasts: Fore each forecast that is produced, the prediction system calculates as well the hindcasts for the same date in the past XX years. It is not clear to me, which hindcasts have been used explicitly by the authors as they mention hindcasts from years (1994-2016). I assume they thus use all hindcast from the forecasts produced during 2017. In addition, on the same line it is written that the hindcasts from 1993-2016 are use, what might just be a typo.

Yes, this was a simple typo. This is now corrected. See L. 138

Another point is the source of predictability. Sources of predictability are from my point of view physical processes and/or connections within the atmospheric/hydrological system. E.g. a source of predictability are sea surface temperature that influence the large scale dynamical patterns, such as for example in the ENSO or NAO, or as a hydrological example, initial conditions of snow or soil moisture can be a source of predictability for river discharges thanks to the memory of the system. I struggle with the terminology used in this manuscript that assigns a seasonal forecasting system as a source of predictability. It is rather the boundary condition, provided by the seasonal forecast system as an input to the hydrological model, that can be seen as the source of predictability. This is already a problem in the title. I suggest the authors to carefully revise the manuscript and the title. The analysis rather aims at determining if seasonal predictions can be used to produce skillfull lake temperature, discharge and ice-off forecasts.

We agree that the SEAS5 forecasting system cannot be seen as a source of predictability, it likely further transfers predictability from e.g., ENSO and NAO, as you mentioned, to our catchment and lake model. We have revised the title as follows:

"Sources of skill in lake temperature, discharge and ice-off seasonal forecasting tools"

We also have carefully rephrased all sentences related to predictability and sources of skill throughout the manuscript, See for example L. 489-491

SEAS5 system does have a resolution of 1° whereas ERA5 has a resolution of 0.25 degree. The Authors mention that ERA5 data was used to bias correct the SEAS5 hindcast. Due to the mismatch in resolution this correction inherently exhibits a downscaling step. Please elaborate what the actual hindcasts are that are used to run the hydrological models and the lake models.

Furthermore, how is the bias correction implemented? Do you use a leave-one-year-out methodology? There is not enough information about this pre-processing step in the manuscript.

This is now better described, see L. 139-150.

More specific comments

Introduction

Line 24-26: two model outputs are compared: seasonal lake hindcasts (forced with SEAS5) and pseudo observations (forced with ERA5). In the next sentence it says "the seasonal lake hindcasts was generally low but higher then SEAS5 climate hindcasts". These sentences are confusing, what is analyzed in the SEAS5 climate hindcasts exactly. Do you mean the skill of the meteorological predictions (SEAS5 outputs) are worse then the skill of the hydrological predictions?

Yes, this is what we meant. We now clarify by splitting the sentence in two and stating more explicitly the point raised here. See L. 25-27

Line 53-55 "Hence, seasonal climate forecasts are usually not the main source of predictability outside the tropics, at least for stream flow (Greuell et al., 2019; Harrigan et al. 2018; Wood et al. 2016)"

I struggle again with this formulation: Sources of skill can be the initial conditions (e.g. snow, soil moisture) or the forcing variables (e.g. temperature forecasts that determine the skill of evapotranspiration). But the seasonal forecast itself cannot, at least from my point of view, be seen as a source of predictability. I suggest that a careful reformulation of this (and similar) sentences throughout the manuscript. In particular, the publications referred to in this sentence can be used as a starting point and the formulations and terminologies used there should be used as well in the current manuscript.

Thank you, we have reformulated throughout the manuscript. See L. 472-473, 539.

Line 70 "...temperature predictions and forecasts" from a meteorological point of view forecasts and predictions are basically synonyms. Thus, I suggest not using both terms to make the manuscript more readable.

Thank you, this was a typo.

Line 72:

"...it doesn't take seasonal climate forecast ensembles i.e. climate data products specifically designed for seasonal forecasting..."this formulation is again confusing for me. The output of seasonal forecasts are seasonal forecasts and not a climate data product. I suggest to avoid such sentences to make the manuscript more clear.

Thank you, corrected.

Line 76 & 77: The authors mention "When forecasting river flow, for example, predictability can originate from two main sources: (i) catchment water stores of initial soil moisture, groundwater, and snowpack, which are directly linked to the water residence time; and (ii) seasonal climate prediction (Greuell et al., 2019)." I agree that there a two main sources (1) the initial conditions and (2) the boundary conditions, i.e. the relevant variables from the driving meteorological forecasts. Again the formulation that "seasonal climate predictions" are a source of predictability is misleading and is to my knowledge not used in literature, as it is a very vague formulation. This makes me feel, without being rude, that the authors should invest more time to familiarize with the commonly used terminology in hydrometeorological forecasting and try to carefully revise the manuscript.

We understand where the confusion was coming from, and we apologize for the lack of precision in our terminology and any impact it might have had on your understanding of this study. This is now corrected. See L. 77-80

Line 80-84: "When dealing with standing water bodies, antecedent conditions are also likely to provide significant predictability, given that the water storage in lakes and reservoirs is large compared to river channels, providing higher inertia. Water residence time is thus expected to exert a strong influence on water flow predictability. Water temperature, on the other hand, is influenced by multiple meteorological variables, e.g., wind, and radiation, in addition to water stores which can affect the source of its predictability."

I do not understand what the authors try to argue, what is meant by "water flow predictability", do you mean discharge of the rivers? Lake level heights? Can you make this sentence clearer?

Yes, we apologize, we meant discharge. This is now corrected

Line 90. I suggest to quickly introduce what ice-off is.

This is now defined at the first mention of ice-off. See L. 74-75

Line 90-91: You mention that you quantified the forecasting skill of each meteorological variable. This is later provided in Table 4. However, this table is hard to understand and does not provide quantified values of the fair RPSS and the ROCSS. Per definition a skill score ranges from one (perfect forecast) to zero (= no skill of the forecast) and negative values indicating that the forecast is less skillful then the reference. Here the authors just indicate if a given skill score is significant for the given variable and tercile. Why don't you provide the actual skill scores and the numbers? And what is your definition of a significant skill? Is everything above 0 a significant skill? Furthermore, the abbreviations of the variables are non-intuitive and seem to be directly from a model output. I suggest using more common/readable abbreviations (eg. rsds = rad or solar radiation). In addition, the variable rlds is only given in this table but nowhere in the manuscript described, what is the meaning of this variable?

To clarify, we have now added a table showing the skill scores and the performance of the lake forecasts. We have removed the variable abbreviations throughout the manuscript, except in tables and figures, where we have opted for more intuitive abbreviations. See Table 4 and Table 5. Also, determination of the significance of the skill scores is now also described at L. 273-274.

Methods:

Line 101: Introduce the definition of ice-off already in the introduction section, this helps the reader to understand from the beginning what you aim at.

Done

Table1: Maybe it helps to add a map, where these reservoirs are located. Are they in complex terrain? In arid or humid climate regions?

We have now added a map in Figure 1, as well as background information on climate type.

Line 109: Climate data

I suggest changing the name of this paragraph to e.g. meteorological input data. Seasonal forecast data is not really a climate dataset, it is in a forecast dataset.

Nice suggestion, thank you. Done.

Line 111: what do you mean with "relatively homogeneous spatial and temporal coverage"? Both datasets, the reanalysis and the forecast data are datasets produced by global earth system models and thus provide a global coverage and are continuous in time as defined by the model integration steps.

Yes, our formulation was confused, this is now clarified. L. 127-129

Line 114-115: Pleas add a bit more detail about the bias correction. Did you use a leave-one-year-out approach? What quantile mapping approach did you use? How do you deal with values above the 99th percentile? Do you use an additive or a multiplicative method? How do you account for intervariable dependencies? Although there are some more information in the supplement of the indexed publication, I think some more information within this manuscript will help the reader to understand what you have done in this study.

Done, see L. 139-150.

Line 116: Here you use the term "impact models" and "impact variables". Are these the water quality models and variables? In the rest of the manuscript the terms "catchment models", "hydrologic models", "lake models"… It makes it hard to follow, when you jump between the different models. I suggest unifying these terms and use one definition throughout the manuscript.

We now stick to "hydrologic catchment model" and "lake physical model" which are then only referred to as "catchment model" and "lake model" for conciseness. In addition, the term "impact variables" which could lead to misunderstandings has been replaced by "output variables" throughout the manuscript.

Line: 125

Here you give more information about the EQM (referred to my comment for line 114-115). I suggest referring to these lines already at line 114. Nevertheless, it is crucial how the bias correction of the seasonal forecasts is done, I suggest adding some more information about how it is actually done. In addition, did you look at the performance of the bias correction? How do skill scores change before/after the correction? In forecast verification, a specific score only shows part of the story, it could be fruitful to look at additional scores and measures, such as e.g. reliability diagrams to get more information about the full ensemble.

We have now added some description of this bias correction step. Unfortunately, we haven't performed any investigation of the performance of the bias correction because of the time and resources allocated to this project. Reliability diagrams are not stable for small sample size and would have required some re-sampling techniques (e.g., Weisheimer and Palmer 2014; https://doi.org/10.1098/rsif.2013.1162) that are out of the scope of this study

Line 127-128:

Here the variable rlds is missing which is shown in Table 4 later. In addition, the abbreviations of the variables are non-intuitiv. I suggest using standard, or more readable abbreviations (e.g. temperature =T, pressure = p or even use there full names in the text). In addition, do you use daily values or lower temporal resolution? Is the daily air temperature a mean daily air temperature? Are these the parameters you really use from SEAS5 and ERA5? Is the air pressure the surface air pressure? You mention wind speed here, but use the u and v wind components, that's completely fine, but then I suggest only mentioning the wind components and not the wind speed, as wind speed is a different variable then (i.e. a combination of) u and v.

As mentioned L. 115, we only use daily resolution for input forcing data. We have also harmonized the abbreviations and variable names throughout the manuscript, avoiding abbreviations in the text. See for example L. 452-453. Note that we also added an index for clarity at the end of the manuscript

Line 130-131: Please describe the data you used and where it was measured? Is the station directly at the lake/reservoir inlet? Is it an official observational station? Who is responsible for the measurements (public agency? Scientific group?)? why are there so many data gaps? How trustful are these observations? What method is used to estimate the discharge? What temperature do you use (daily mean, daily max, daily min)? There is no need to include all this information, but at least a reference about how trustworthy the measurements are would be necessary.

Done, we have added a description of the observations. L. 158-170

Line 142: Here you use the variable abbreviations, as mentioned before, I suggest using more intuitive abbreviations or writing the full names of the variables, to improve the readability of the manuscript.

Done

Line 143-145:

The authors mention that all hydrological models have been validated against observations. Do you have any publication you can refer to? What is the performance in terms of NSE for the calibration? Were the models calibrated for the same time period/locations? Was the same observational dataset used? Please give some more information.

All catchment and lake models were calibrated against local observations (this is now explicit at L. 158-159 and 242-243). We have also added a table describing the calibration/validation time periods and verification statistics (Table 4), and refer to Table S2 for additional statistics by season.

Line 152: Maybe it is worth to introduce these performance measures in a separate chapter, e.g. together with the skill scores.

We believe this is the best place to first introduce these performance measures, since we describe the models used in the workflow. However, note that we refer to them again later together with the other skill scores when we describe the different verification steps we have performed (Table 2 and section 2.2.1).

158-159: "Most common statistical goodness-of-fit parameters, e.g., Kling-Gupta efficiency (KGE), NSE and RMSE, for hydrological and lake modeling were calculated."

What is the results in terms of these scores? Specifically mention the numbers, otherwise this sentence is obsolete. In addition, I struggle with the formulation "goodness-of-fit paramters", KGE, NSE, RMSE are performance criteria or scores to evaluate the model performance. I suggest revising this formulation and use more commonly used formulations from the field of forecast verification.

Performance measures are now given in Table 4. And we have replaced the expression "goodness-of-fit" by "verification statistics".

Line 167-168: What do you mean with " total predictions skill" and what observations have been used to assess the performance? Do you here refer to the hydrological part I assume? I suggest a more careful formulation to better discriminated between the meteorological and the hydrological parts.

Total prediction skill also accounts for error or bias originating from the model itself. We now clarify as follows. L. 226-228

Line 173-174: "Over the initialization month, the 25 members of SEAS5 progressively diverge from ERA5 to their respective SEAS5 member."

How is this transition done? In the text you say initialization month, whereas in Figure 2 you mention transition month, I suggest unifying the terminology to avoid confusion.

We apologize for this confusion again. We have unified the terminology regarding the first lead month (M0) and have updated the figure (now Fig. 3), following Greuell et al. (2019), see also our response to your last comment.

Line 174-175:

"Model outputs for the final 3 months, i.e., the target season, were selected and used to calculate the probabilistic forecasts of seasonal summary statistics."

This sentence illustrates what I mean with imprecise language. "The model outputs for the final three months" are already the probabilistic forecasts and for these forecasts, the scores are calculated. I suggest carefully revising this (and similar) formulations. In addition: Does it mean that you only use 3 month lead times? The seasonal forecasts provide predictions up to 13 moths, why do you use only 3 months lead time? I think this should be mentioned in the very beginning of the manuscript (and already in the abstract), as it is important to know what lead time horizon you are focusing on.

Yes, we only use 3-month lead time. This is now mentioned in the abstract and introduction. See L. 24, L. 91 and 94

In addition, it is not clear to me how you do your verification. Is it based on daily values? Weekly averages? Monthly averages? The full 3 month period? Do you do a lead time dependent verification as well?

Verification is based on the 3-month seasonal averages. We only focus on the 3-month lead time, we don't do any lead time dependent verification because of constrain on time and resources. This is now clarified L. 245-249.

Line 191:

"Both skill scores are expressed as relative to a reference forecast, i.e., climatology."

It would be worth to more carefully explain the concept of skill scores. A Skill Score is a comparison of two scores, one calculate for the forecast of interest and one for a reference forecast. Thus again this sentence is somehow imprecise and should be revised. Hence, for calculating the skill scores, what climatology did you use? I assume you used the climatology based on the pseudo observation experiments, is that correct? If so, please mention this in the manuscript

We agree that this sentence was imprecise, we have removed it and skill scores are introduced earlier, see L. 256-260

Line 194-197: I do not understand the concept of "windows of opportunity". Can you better explain how these windows of opportunity were selected? And why only these windows were taken into account for the evaluation?

We describe it better and at multiple places throughout the manuscript to ensure clarity. See e.g., L. 99-104, 274-276

Note that all forecasts went through the verification steps described in Table 2 however, for the conciseness of the paper, we only selected sites that had the most hindcasts with significant performance (i.e., windows of opportunity) for the sensitivity analyses (L. 283 and L.296-297).

Table2 and full paragraph:

I struggle with understanding your procedure. Is the "Evaluation data" in table 2 the data used to compute the reference forecast for the skill scores? If so, please change the name from evaluation data to reference forecast data or similar. For evaluation of a forecast, you anyway need explicit data at the same temporal resolution as your forecasts (being real or pseudo observations). If you use a climatology (based on real or pseudo observations) as your reference forecast, how is the climatology constructed? How long is the timeseries that you take into account and how many data gaps do you allow for constructing your climatology based on real observations. If you mention that there are large gaps, it might be difficult to construct a reliable climatology.

We now describe step by step our verification procedure which included three steps. We also explicitly state at which temporal resolution we have performed those verifications, and against which reference forecast (L.241-249). We have updated Table 2 to make it clearer as the reviewer suggested.

Line 205-207: these two sentences are a repetition. I suggest reformulating it.

Done. See L. 287-290.

Line 200-227: I struggle a lot to follow the explanation. First of all what is
ROCSS**s**/ ROCSS**w**/ ROCSS**w+t**

Refer to table 3 early in this paragraph, then it becomes already a bit clearer!

Line 209-230: Again the formulation makes it very difficult to follow and the abbreviations
further reduce the comprehensibility. I suggest to reformulate this paragraph and emphasize on
better understanding.

Sorry for the intricate formulations here. We have reformulated this paragraph and now
explicitly describe how we performed the sensitivity analyses and how the various ROCSS values
were calculated. See L. 281-286. Note that we already referred to Table 3 in the first version of
the MS, but we now do it in a more explicit way at L. 281. Note also that $ROCSS_S$, $ROCSS_W$ and
$ROCSS_{w+M0}$ (formerly $ROCSS_{w+T}$) are now explicitly described L. 299-303.

Maybe it would help to choose simpler or more descriptive titles for the paragraphs. E.g. 2.2.2
could be sensitivity analysis to initial conditions and meteorological forcing (input periods); and
2.2.3 Sensitivity to individual input variables.

Thank you, we followed your advice and changed the title of those sections.

Line 231: "..each Lake_PO variable" do you mean with each output variable of the Lake_PO
experiment? State that explicitly, it will be much easier to understand!

Sorry for the confusion, done. L. 316

Line 240-244:

Is this now based on your analysis? Or are these theoretical considerations? This is not entirely
clear to me. In addition, the formulation is a bit misleading, does it mean that the variables that
are retained are the variables that are used from seasonal meteorological predictions to run the
hydrological models?

Admittedly, this was not well described. We have included a description of the various heat
fluxes L. 207-216. We have now included an assessment of the importance of the various
discrete heat fluxes to the lake heat budget in the supplementary material and refer to it L. 330.

Results

Line 251: you refer to table S2 in the supplement. I suggest adding part of the table in the main
manuscript and refer to "additional information" in the supplement.

We have added a table in the manuscript showing the calibration and validation verification
statistics (Table 4) and also refer to Table S2. See L. 334-337

Line 262: Here you mention fair RPSS, which accounts for a limited number of ensemble size.
However, this is not introduced in the methodology section, there you should at least mention
that the fair (or debiased) RPSS is used and what it accounts for.

Again, what do you mean with "significant fair RPSS"? When do you assume an RPSS to be significant?  How do you test the significance of a skill score?

Fair RPSS is now described in section 2.2.1 at L. 270-274. And we have replaced RPSS with FRPSS and have removed any redundant expressions.

Line 269-270:

"Only 0 to 10% of the SEAS5 climate 270 hindcasts are skillful, on average".

I struggle understanding this result. The RPSS (or any score) is usually determine based on a large sample forecasts (or hindcasts). Of course, for an individual forecast this might be poor but the full picture of forecast performance can only be revealed when the Scores for many issued forecasts are investigated.

I think here you should add a figure with the results where the reader can see how the hindcast performance actually is. From the text and the table alone, it is rather difficult to follow your argumentation. In addition, what is the definition of a skillful hindcast in your context? Is every hindcast with a ROCSS>0.5 skillful or do you use other thresholds? It would be crucial to mention the numbers at least once in the results section as well.

We have added a table showing the performance of the Lake_F by displaying the ROCSS_original values by output variable, season and tercile (now Table 5). Please also see added text L. 345-348. The threshold for skillful hindcast is described in the methods section, see L. 273-277, and mentioned in the results section, see L. 348. Note that we rephrased the sentenced (originally at L. 269-270) at L. 361-362

Line 278-280: How many seasons are discarded due to missing observations? Please indicate the exact numbers such that the reader knows how many samples (hindcasts) are actually used for the analysis. Or at least refere to the table where the numbers are listed.

We now refer to Table 5.

Line 283-286: You mention that the ROCSS does not capture the same as the "goodness of fit statistics" do. This is by definition the case as they do not look at the same properties of a forecast. Therefore, for forecast verification multiple scores should be taken into account to properly assess the forecast performance. Can you rephrase the sentence to make it clearer what you actually mean?

These statements were deleted, as the reviewer pointed out, this is by definition the case. Note that we focus of the ROCSS in this study because this is how we have defined our windows of opportunity.

Table 4

Again general comments: abbreviations are non intuitive. FRPSS is not introduced before. I do not get the message of this table. I would prefer to see the skill scores (e.g. as boxplots) over different seasons for the variables. It is not clear to me what temporal aggregation is the baseline of this analysis.

Thank you, we have defined the abbreviations in the table caption. Temporal aggregation is now described in the method section L. 243-245, as well as FRPSS L. 270-274. Skill scores are now presented in Table 5. Note that Table 4 is now Table 6.

Table 5

Here you use abbreviations for the seasons (SP, WI, AU, SU), although it can be inferred I suggest to avoid introducing additional abbreviations here. Again, how do you determine the significance of the ROCSS?

Thank you, we have replaced abbreviations with full name. Note that significance of the ROCSS is now described in section 2.2.1 at L. 273.

Line 305:

Fig. 31 should be Fig. 3

In fact, this is now Fig. 4 panel "l". This is now explicitly stated.

Can you elaborate how the ROCSS is determined, what exact values are taken into account? Do you use daily values to calculate the scores or weekly/seasonal aggregated values? This is still unclear after reading the manuscript.

This is now described in section 2.2.1.

Fig 3: it is confusing that in the plot description and the text you mention ROCSSs etc. but on the x axis S-SA W-Sa etc are displayed. I suggest to unify all and make the plot more readable.

Thank you for spotting this. This is now corrected.

Line 320-322:

Maybe because I do not understand what the windows of opportunity are, I do not get the message here. I suggest to more explicitly formulate what the impact is of changing the initial conditions and the forecast input. The result indicates that it is not worth using seasonal forecasts at all, which is hard to believe. Please elaborate such that the message gets clearer.

Now that the concept of windows of opportunity is defined earlier and repeated at multiple places throughout the MS, this section should be clearer. Also note that we have recalled the impact of changing the initial conditions and forcing data within the sensitivity analyses, see L. 403-405.

Line 323: again the title does not reveal to me what will be considered in this paragraph. I suggest to use a more appropriate title for this paragraph.

Thank you, we have updated it. L. 419

Figure 4: It is hard to follow what is shown here. What is the relative sensitivity. Maybe it helps if you refer to the exact paragraph number in the label of the figure. In addition, is there a reason

why you use a color coding and a size coding? I suggest either using color or size, otherwise it seems that multiple aspects are coded.

We have updated the caption to make this figure (now Fig. 5) more understandable at L. 448-450. And you are right, there is no reason for both color and size coding. Hence, we have removed the color coding.

Line 347-348: "Hence, a significant fraction of predictability is originating from the SEAS5 dataset although the largest source remains ERA5 data over the warm-up". This sentence illustrates what I think makes the manuscript complicated to follow. The reader himself must make the connection what this means. If I am correct, it shows that the initial conditions are more important than the driving meteorological predictions. Is this correct? It would make the manuscript much more readable if you directly refer to formulations what it actually means in addition to just give such "cryptic" explanations.

This is now correct at l. 470-471.

366-368: … sources of seasonal water quality skill… You mean water quality forecast skill, water quality itself does not really have skill, does it?

Yes, sorry again for this imprecise formulation. We have reformulated, see L. 485

Line 394-395:

Literature on streamflow hindcasts broadly shows that beyond the transition month, climatology-driven hindcasts are typically 395 more skillful than hindcasts driven by seasonal climate predictions (Arnal et al., 2018; Bazile et al., 2017; Greuell et al., 2019).

This is a misleading interpretation, when you say "climatology driven hindcasts". In all three publications a well established ESP (ensemble streamflow prediction) approach is used. This can be seen as a climatology driven hindcast, but for a scientific publication I would expect to have a clearer formulation. In addition, in these papers there is no transition month mentioned, a concept I do not understand. All papers mention the first lead time month. What exactly is the transition month in your analysis?

We have rephrased this sentence, see L. 512-514.

Note also that we replaced the "transition month" with "first lead month M0", in agreement with Greuell et al., 2019, i.e., the month following the date on which the forecast would have been issued. This clarification was implemented throughout the manuscript.

---

## Referee Report (RR1)

**Original title: Inertia and seasonal climate prediction as sources of skill in lake temperature, discharge and ice-off forecasting tools**

**Revised title: Sources of skill in lake temperature, discharge and ice-off seasonal forecasting tools**

Editor's and reviewers' comments are underlined for clarity. Line numbers refer to line in the revised (clean) version of the manuscript, if not specified otherwise.

**Editor:**

Both reviewers have indicated that this is an important topic, but they have also raised important issues requiring considerable work. Thank you for responding to both reviewers. However, your responses remain very descriptive, and it is challenging to assess whether the changes will sufficiently address the issues raised by the reviewers. Together with the revised manuscript, please provide a detailed point-by-point responses to all reviewers' comments.

Thank you for your time and consideration. We provide below a point by point response to your comments and to each reviewer's comments.

I have finally highlighted two points that I believe need special attention when working on the revised manuscript: * clarity. Both reviewers have stressed that the clarity, languages and terminologies have not reached a publication level. See specifically the detailed comments from RC2.

We have taken special care in rephrasing many imprecise sentences and have tried our best to use the most precise terminology. We believe the manuscript has been improved significantly.

* Retention time. Given the short retention time of the selected lakes, the advective heat might be highly important (see RC1). Is it realistic to not take into account the heat change from the throughflow? See for instance: Schmid, M., & Read, J. (2022). Heat budget of lakes. In T. Mehner & K. Tockner (Eds.), Vol. 1. Encyclopedia of inland waters (pp. 467-473). https://doi.org/10.1016/B978-0-12-819166-8.00011-6 Råman Vinnå, L., Wüest, A., Zappa, M., Fink, G., & Bouffard, D. (2018). Tributaries affect the thermal response of lakes to climate change. Hydrology and Earth System Sciences, 22(1), 31-51. https://doi.org/10.5194/hess-22-31-2018

We had overlooked this in the previous version of the manuscript, but indeed, we have taken inflow water temperature into account in the heat bugdets. A side analysis, for review only, shows that the assumption the reviewer 1 raised doesn't hold. See our response to his comment for additional details.

**Reviewer 1:**

This article treats the important and so-far under developed field of seasonal forecasts (here 4 months into the future) of lake and drainage area properties including water temperature, ice-off, and river discharge. The authors combined two lake models (simulating surface and bottom temperature and ice cover, two lakes each) with four Hydrologic models (simulating discharge, one drainage area each). The method was applied at four lake-river system located in Norway, Spain, Australia, and Germany. Modeled systems include lakes spanning 19 o 60 meters depth and with a retention time from 0,2 to 1,1 years.

The coupled model setup was calibrated towards measurements (lake temperature and river discharge) and forced with reanalyzed data from ERA5, I would define this as a general circulation model (GCM). Hydrological-lake model performance was evaluates with KGE, NSE and RSM. Thereafter calibrated models was spun-up during one year and forecasted discharged and lake surface and bottom temperature during four months (one month initialization) spanning 13 years (1993-2016). Future forcing comes from 25 forecasts from the global forecasting tool SEAS5. SEAS5 was bias corrected and downscaled (grid adjustment) towards ERA5 to enable comparison.

The correctness of the forecasts (Lake_F) was evaluated trough a sensitivity analysis, comparison of Lake_F towards in-situ measurements and towards daily pseudo-observations (Lake_PO, daily output from the coupled hydrological-lake model setup forced with ERA5). The end product of this manuscript consist in an evaluation (sensitivity analysis), of forecasting correctness for each river-lake system and a evaluation of forcing parameters influence on forecasts.

The manuscript show potential but is lacking in some areas which I list hereunder.
Thank you for this thorough and helpful review. We have carefully rephrased many imprecise sentences and have streamlined the manuscript. We also have added details on the case-study sites, observation data, our modelling procedure and how we dealt with inflow water temperature. Below are our responses for each of your specific comments.

The manuscript and language has improved but as my replies show hereunder it still needs to be made clearer for the reader.

**Chosen drainage areas and lakes**

The authors put forward that seasonal predictions work best next to the equator and worsen with increased latitude (line 50 to 58). Yet, no system was chosen in this region, Spain being the closest. The manuscript could still benefit from an analysis of latitudinal effects for the used forecasting method to improve forecasting towards the North/South pole.
Unfortunately, this is out of the scope of this study. We only have looked at four case-studies outside the tropics to investigate any opportunities and have "ready to go" workflow when seasonal meteorological predictions improve significantly. See L. 126-129

My comment did not entail an extended survey adding more systems, but rather if latitudinal effects could be distinguished in the present setup from the lakes and rivers you used. Furthermore skill is related to each individual lake in the manuscript. I could not find details regarding the impact of each river model (SimplyQ, mHM, GR..) nor lake models (GOTM, GLM) on the forecasting results. This need clarification in the manuscript, at least in the form of a discussion in order for the reader to correctly interpret your results.

Additionally, the river-lake systems chosen contain lakes with very short retention time, i.e. big impact of rivers on water constituents, including temperature. The model method used include the effect of changing lake volume, but not the effect of heat being transferred into the lakes by upstream drainage area (input temperature I could not find). Thereby it is reasonable to assume that the lake models (through calibration) had a better connection between surface and deep waters than is the in-situ case.

Could this show up in your analysis of forcing parameter importance ("Tracing of forecasting skill" section 3,4, Fig. 4)? This needs to be addressed/analyzed since you link forcing to lake processes, which in fact could be caused by upstream heat fluxes in the drainage area and not in the lakes themselves. Admittedly, this was not clear in the manuscript. Nonetheless, we did take water temperature of the inflows into account. This is now announced early in the methods section, see L. 117-118 and described in detail at L. 207-216.

As a side-analysis, for review only, we have looked into the correlation coefficients between lake surface and bottom temperature for observations and pseudo-observations (modelled temperature with ERA5 data as forcing data). This analysis revealed that the correlation for pseudo-observations was not necessarily higher than for observations discrediting the reviewer's assumption that "lake models (through calibration) had a better connection between surface and deep waters than is the in-situ case" and giving further confidence that the heat lake budget is robust. In fact, the correlation coefficients for both observations and pseudo-observations ranged between 0.22 and 0.96 depending on the season and case-study. In given seasons, the observations even showed higher connections between surface and bottom water than pseudo-observations.

Now, we don't believe that adding inflow temperature in our analysis in section 3.4 would add any significant insight since inflow temperature is largely based on air temperature which is already accounted for. In order to avoid any redundancy in our analysis and following principles of parsimony. Note that we have added an assessment of the various lake heat fluxes in the supplementary information and refer to it at L. 330.

My concern was that river temperature looked to be excluded from the simulations, now it is apparent that it was included so I am fine with the authors additions to the manuscript. I looked at the annual heat fluxes in the S.I., the through flow was between 5 and 12% of the total heat flux and at 3 out of 4 lakes it came at 4th place (e.i. in order largest to smallest: short-, long-wave, latent, throughflow and sensible heat fluxes).

**Data**

This manuscript use ERA5 reanalysis as a stand in for in-situ measurements. Why is this, due to large spatial extent of drainage areas? If possible, show how this influence your modelling locally, or refer to documents where the reader can find this comparison between ERA5 and in-situ measurements, in best case for the regions being analyzed.

We forced our models with ERA5 meteorological data to ensure that our workflows were comparable between each case-study and future transferability of these workflows. We clarify this now at L. 126-129. In addition, weather observations covering the whole range of variables needed to force our models were not available over the whole period from 1994 to 2016. Yes, this has likely influenced our modelling locally. Note however that, for ground truthing, we include a forecast verification step compared to a reference forecast based on observations which is not often included in forecasting studies (Table 7).

This is good, but why do you have data gaps in verification statistics in table 7? I.e. you have $ROCSS_{original}$ which require Lake_PO and Lake_F values but do not show verification statistics for Lake_PO.

**Clarity**

The manuscript could benefit greatly from an index defining the many acronyms used, as well as improved description of tables and figures. Ex Table 4 and 5 is hard to understand.

Thank you. We have added an index at the end of the manuscript. L. 577.

Good, but have a look so that you haven't missed any acronyms, I found $R^2$.

Furthermore, I could not find/understand if the drainage area and lake models are coupled in time (run simultaneously), or if the drainage area models where run in advance to provide discharge for the lake models.

This is now clarified L. 114-118

Good

**The language**

Certain words in the manuscript cause some confusion. Bellow I have stated some that might need to change

Skill – is associated with people. A fast car (a tool) has no skill it has performance, the driver on the over hand has skill. That said, I know skill is used more commonly to describe models (tools) in meteorology than hydrology. So I suggest that you define what you mean by skill if you want to keep this formulation. Skill is now defined twice in the introduction (L. 53 and 88-89) and repeated in the results (L. 347)

Good

Climate & climate prediction – studies involving effect of climate focus on longer time periods (>30 years) than what is the focus in this study (<1.5 years). Both SEAE5 and ERA5 comes from global GCM models, which could be used for climate studies. But in the context of this manuscript I do not think this is the right phrase describing the models you used.

Agreed, there was some confusion between climate and meteorological variables. This is now clarified throughout the manuscript. We rather refer to "seasonal meteorological forecasts/predictions" and mention explicitly the variables, when possible.

Good

Hindcasts – is usually used in the setting of running models with data from past events, close to reanalyze with the aim to improve said models. Here this word is used in combination with SEAE5 forecast simulations. The authors have adjusted these to ERA5 (real data proxy) but the intention is still to use SEAE5 as forecasting forcing. Therefor consider other alternatives in the manuscript, or define this word in the context of your manuscript.

Hindcast is now properly defined in the introduction, see L. 93

Adjust this sentence to read clearer, otherwise acceptable

Water quality – for drinking water and the biosphere, temperature is considered an important water quality parameter. Here we do not look at lakes and rivers in this sense, water quality one would assume here to entail dissolved constituents (nutrients, oxygen…). To avoid misunderstanding, consider using something else.

Agreed, we have replaced water quality by relevant alternatives throughout the manuscript, e.g., water temperature, lake.

Good

Line 19 : "as previously presented". Avoid need for reference in abstract.

We have removed this part from the sentence.

Good

Line 67 : Consider adding the following reference https://doi.org/10.1016/j.watres.2020.115529

Thank you!

Good

Line 72 to 74: partly untrue, air2water can run perfectly with seasonal forcing as you do here (only air temperature as forcing), and ice-off is currently available indirectly.

The sentence has been updated. L. 71-72

This air2water model is constrained to stop at 0 °C, i.e. ice-on and ice-off is indirectly modelled while other ice processes such as ice thickness is omitted. Adjust text accordingly.

Section 2,1,1 : The reader do not know where the lakes and drainage area (rivers) under investigation is situated. Add a map showing the global location and regional extent of each drainage area-lake system

(rivers and lakes). Additionally add these system details (names, stations, etc.) where appropriate, ex. Table S1.

We have added a map (new Fig. 1) and detailed catchment maps are given by Jackson-Blake et al. 2022 as well as more background information on the case-studies. We now refer the readers to this study. See L. 110-111.

Good, but I read through the following paper and could not find the detailed catchment maps referred to in Figure 1. Add spatial catchment extent of upstream rivers that was modelled. Furthermore, using a figure already published requires permission from original creator, now since the author list below is similar to the list in this manuscript I leave it up to the editor to decide if a written permission to use the adjusted figure is required or not.

Jackson-Blake, L. A., Clayer, F., de Eyto, E., French, A. S., Frías, M. D., Mercado-Bettín, D., Moore, T., Puértolas, L., Poole, R., Rinke, K., Shikhani, M., van der Linden, L., & Marcé, R. (2022). Opportunities for seasonal forecasting to support water management outside the tropics. Hydrology and Earth System Sciences, 26(5), 1389–1406. https://doi.org/10.5194/hess-26-1389-2022

Line 111 or 112 : add reference: SEAS5: the new ECMWF seasonal forecast system. Stephanie J. Johnson, (2019), https://doi.org/10.5194/gmd-12-1087-2019

Done

Good

Line122 to 123 : „Climate data where downloaded….". What do you mean in this sentence, ERA5 and/or SEAS5?

This sentence has been removed and we now provided much more details on ERA5 and SEAS5 data pre-processing steps, as requested by the other reviewer. See L. 139-150

Good

Line 139 ")" missing

Thank you, Corrected.

Good

Line 156 to 157 : Add details (equations and ex. RMSE) of this linear regression between in- versus outflow.

This is now described in more details L. 195-206 and in the supplementary material.

Good

Figure 2. consider showing mean of SEAS5 predictions and ERA5 at the same time (i.e. continue black lines into transition and target season).

We already show the mean of SEAS5 predictions, adding a line would overload the figure (now Fig. 3). Besides, the main concept of this figure is to illustrate the input data used to force the model. ERA5 data was used only over the warm-up period.

Good

Line 185 : RPSS looks to be missing from table 2 and table 3.

FRPSS was not considered for formal forecast verification because it doesn't allow to distinguish forecast performance for a given tercile. With these low performance lake forecasts that we have reported, this FRPSS didn't seem very useful to us. However, we still include it in Table 6 for reference.

Good

Line 235 to 238 : something is missing here, hysteresis should make linear relationship between ex. air temperature and water temperature rather bad. Describe how good these linear fits were (in appendix). And/or show with figure and improve explanation.

Pearson partial correlation coefficients (PPCC) are calculated from seasonal means, and not daily values, which likely yielded much cleaner correlation than expected from daily values. This point is made clear now see L. 321.

Good

Line 243 the reader are not familiar with the contributions of local heat fluxes at the chosen locations. Before disregarding for example cloud cover from the analysis, show the reader in numbers (or preferably figure as appendix) for each lake the seasonal heat budget contributions. I.e. uptake and emission of infrared longwave radiation, evaporation + condensation, sensible heat flux and uptake of surface downward solar radiation. Throughflow you only have the outflow (at some lakes?) since inflow temperature is missing.

This background information is now described L. 207-211 and we refer to the supplementary information L. 330 for further details.

Good

Line 250 : RMSE not consistent with RMSE/sd in Table S2. What is RMSE/sd? Use the same in text as in Table S2.

This is now clarified, these first performance measures are for the whole year (now in Table 4) while Table S2 shows measures by season. RMSE/sd is now defined in the caption of Table S2.

Good

Table 5. move description of asterisk under table and improve the site representation. Now you can not see what belongs to which system. And define the season duration.

Done.

Good

Figure 3 and 4 : missing Germany and Australia, add or explain.

Note that we precise that the four probabilistic sensitivity analyses, S-SA, W-SA, W+M0-SA and OAT-SA were only performed at the sites in Spain and Norway because of the significant resources needed to execute these hindcast experiments (see L. 296-298). This point is now repeated in the figure captions (now Fig. 4 and 5).

Good

Figure 4 : Something do not add up in your analysis. Top row for Spain – Bottom temperature, and Norway - Surface temperature appear to be to large compared to the individual season values taken together. I.e. if the impact is small most seasons, I do not see how it could be much larger on an annual basis.

We agree with you that it can appear quite surprising that the sensitivity calculated on an annual basis is much larger that over each single season. However, Norway's climate is subjected to strong seasonality which can be well captured over annual scales, but still have low correlation for a given season. To make this point clearer, we added a sentence in the figure caption (now Fig. 5). See L. 451-454.

I don't follow you reasoning here. One would expect that surface temperature (ST) in summer is sensitive to for example short wave radiation (SWR), but during a complete year this sensitivity should be lessened by decreased importance of SWR for ST. If the seasonal signal intrude into multiple time frames (summer, autumn etc.) I would expect that the dependency is carried over from time window to time window and thereby show up for each season in your analysis.

Furthermore from the new figure (Fig. 5) it is now more clear what the size of the circles represent. Yet I could not find the definition for $R^2$, is it the coefficient of determination? If so, then there is an error in the figure and on line 319 since a high coefficient of determination (towards 1) show good correlation, thereby $1-R^2$ would give small and not large circles for more influential input variables. Is this so?

Figure 5 : Why so many data gaps? Consider showing seasons where significance is worse (higher) but clearly state which significance level you trust.

This figure has been updated to show all correlation coefficients, even those that are not significant, and we now precise the significance level we trust in the figure caption (now Fig. 6) at L. 458-459.

Good, but use different color range for PPCC, try for example white color around 0.

Line 468 : add author contributions. Who did what?

Done, see L. 626-629

Good

Additional points

It need to be more clear what is included in the lower, middle and upper terciles used throughout the manuscript. As an example it is now hard to understand Table 5, which show that your predictions can be better for lower and upper terciles compared to middle. I understand it as terciles composing a diversion of the model output so that extremes end up in the lower and upper terciles. With this understanding it now looks like from Table 5 that you are better at predicting extremes rather than the majority of events occurring in your system.

Add the following from Supplement to the main manuscript

You can find all the codes and data files related to this manuscript at: https://github.com/NIVANorge/seasonal_forecasting_watexr

Figure 4: Extend y-range of plots, now boxplots are cut-off by the x-axis

Line 239: Incorrect title number labeling

---

## Author Response (AR2)

**Second Responses to comments on Manuscript HESS-2022-312 | Research article by Clayer et al.**
**Our responses are in blue below. Line numbers in blue refer to the track-change version of the revised manuscript (R2).**

**Reviewer 1:**

The manuscript and language has improved but as my replies show hereunder it still needs to be made clearer for the reader.
**Thank you, we provide below a point-by-point response to your comments.**

**Chosen drainage areas and lakes**
The authors put forward that seasonal predictions work best next to the equator and worsen with increased latitude (line 50 to 58). Yet, no system was chosen in this region, Spain being the closest. The manuscript could still benefit from an analysis of latitudinal effects for the used forecasting method to improve forecasting towards the North/South pole.
Unfortunately, this is out of the scope of this study. We only have looked at four case-studies outside the tropics to investigate any opportunities and have "ready to go" workflow when seasonal meteorological predictions improve significantly. See L. 126-129
My comment did not entail an extended survey adding more systems, but rather if latitudinal effects could be distinguished in the present setup from the lakes and rivers you used.
Furthermore skill is related to each individual lake in the manuscript. I could not find details regarding the impact of each river model (SimplyQ, mHM, GR..) nor lake models (GOTM, GLM) on the forecasting results. This need clarification in the manuscript, at least in the form of a discussion in order for the reader to correctly interpret your results.
**The number of sites included in our study is preventing a detail analysis regarding the impact of specific river and lake models on forecasting results. Regarding latitudinal effects, the fact that predictive skills were the highest at the northernmost site (for meteorological and lake hindcasts) suggests that predictive skill does not necessarily decrease as we move away from the tropics, and highlights some opportunities. Following the reviewer advice to make those points clearer, we have added some text in the discussion, see L. 545-555.**

Additionally, the river-lake systems chosen contain lakes with very short retention time, i.e. big impact of rivers on water constituents, including temperature. The model method used include the effect of changing lake volume, but not the effect of heat being transferred into the lakes by upstream drainage area (input temperature I could not find). Thereby it is reasonable to assume that the lake models (through calibration) had a better connection between surface and deep waters than is the in-situ case.
Could this show up in your analysis of forcing parameter importance ("Tracing of forecasting skill" section 3,4, Fig. 4)? This needs to be addressed/analyzed since you link forcing to lake processes, which in fact could be caused by upstream heat fluxes in the drainage area and not in the lakes themselves.
Admittedly, this was not clear in the manuscript. Nonetheless, we did take water temperature of the inflows into account. This is now announced early in the methods section, see L. 117-118 and described in detail at L. 207-216.
As a side-analysis, for review only, we have looked into the correlation coefficients between lake surface and bottom temperature for observations and pseudo-observations (modelled temperature with ERA5 data as forcing data). This analysis revealed that the correlation for pseudo-observations was not necessarily higher than for observations discrediting the reviewer's assumption that "lake

models (through calibration) had a better connection between surface and deep waters than is the in-situ case" and giving further confidence that the heat lake budget is robust. In fact, the correlation coefficients for both observations and pseudo-observations ranged between 0.22 and 0.96 depending on the season and case-study. In given seasons, the observations even showed higher connections between surface and bottom water than pseudo-observations.

Now, we don't believe that adding inflow temperature in our analysis in section 3.4 would add any significant insight since inflow temperature is largely based on air temperature which is already accounted for. In order to avoid any redundancy in our analysis and following principles of parsimony. Note that we have added an assessment of the various lake heat fluxes in the supplementary information and refer to it at L. 330.

My concern was that river temperature looked to be excluded from the simulations, now it is apparent that it was included so I am fine with the authors additions to the manuscript. I looked at the annual heat fluxes in the S.I., the through flow was between 5 and 12% of the total heat flux and at 3 out of 4 lakes it came at 4th place (e.i. in order largest to smallest: short-, long-wave, latent, throughflow and sensible heat fluxes).

**Thank you, for your thorough assessment.**

**Data**

This manuscript use ERA5 reanalysis as a stand in for in-situ measurements. Why is this, due to large spatial extent of drainage areas? If possible, show how this influence your modelling locally, or refer to documents where the reader can find this comparison between ERA5 and in-situ measurements, in best case for the regions being analyzed.

We forced our models with ERA5 meteorological data to ensure that our workflows were comparable between each case-study and future transferability of these workflows. We clarify this now at L. 126-129. In addition, weather observations covering the whole range of variables needed to force our models were not available over the whole period from 1994 to 2016. Yes, this has likely influenced our modelling locally. Note however that, for ground truthing, we include a forecast verification step compared to a reference forecast based on observations which is not often included in forecasting studies (Table 7).

This is good, but why do you have data gaps in verification statistics in table 7? I.e. you have $ROCSS_{original}$ which require Lake_PO and Lake_F values but do not show verification statistics for Lake_PO.

**There are data gaps in the verification statistics (NSE, R2, RMSE, RMSE/sd, bias) for Lake_PO seasonal means because these are calculated by comparing Lake_PO to observations and observations are not covering all seasons. To make sure this point is clear, we added some precisions in the caption (see L. 401).**

**Clarity**

The manuscript could benefit greatly from an index defining the many acronyms used, as well as improved description of tables and figures. Ex Table 4 and 5 is hard to understand.

Thank you. We have added an index at the end of the manuscript. L. 577.

Good, but have a look so that you haven't missed any acronyms, I found $R_2$.

**Agreed, R2 was missing. We added it and check for any other missing acronyms.**

Furthermore, I could not find/understand if the drainage area and lake models are coupled in time (run simultaneously), or if the drainage area models where run in advance to provide discharge for the lake models.

This is now clarified L. 114-118

Good

**The language**
Certain words in the manuscript cause some confusion. Bellow I have stated some that might need to change

Skill – is associated with people. A fast car (a tool) has no skill it has performance, the driver on the over hand has skill. That said, I know skill is used more commonly to describe models (tools) in meteorology than hydrology. So I suggest that you define what you mean by skill if you want to keep this formulation.

Skill is now defined twice in the introduction (L. 53 and 88-89) and repeated in the results (L. 347)

Good

Climate & climate prediction – studies involving effect of climate focus on longer time periods (>30 years) than what is the focus in this study (<1.5 years). Both SEAE5 and ERA5 comes from global GCM models, which could be used for climate studies. But in the context of this manuscript I do not think this is the right phrase describing the models you used.

Agreed, there was some confusion between climate and meteorological variables. This is now clarified throughout the manuscript. We rather refer to "seasonal meteorological forecasts/predictions" and mention explicitly the variables, when possible.

Good

Hindcasts – is usually used in the setting of running models with data from past events, close to reanalyze with the aim to improve said models. Here this word is used in combination with SEAE5 forecast simulations. The authors have adjusted these to ERA5 (real data proxy) but the intention is still to use SEAE5 as forecasting forcing. Therefor consider other alternatives in the manuscript, or define this word in the context of your manuscript.

Hindcast is now properly defined in the introduction, see L. 93

Adjust this sentence to read clearer, otherwise acceptable

**Adjusted see L. 95-96**

Water quality – for drinking water and the biosphere, temperature is considered an important water quality parameter. Here we do not look at lakes and rivers in this sense, water quality one would assume here to entail dissolved constituents (nutrients, oxygen…). To avoid misunderstanding, consider using something else.

Agreed, we have replaced water quality by relevant alternatives throughout the manuscript, e.g., water temperature, lake.

Good

Line 19 : "as previously presented". Avoid need for reference in abstract.
We have removed this part from the sentence.

Good

Line 67 : Consider adding the following reference https://doi.org/10.1016/j.watres.2020.115529
Thank you!

Good

Line 72 to 74: partly untrue, air2water can run perfectly with seasonal forcing as you do here (only air temperature as forcing), and ice-off is currently available indirectly.
The sentence has been updated. L. 71-72

This air2water model is constrained to stop at 0 °C, i.e. ice-on and ice-off is indirectly modelled while other ice processes such as ice thickness is omitted. Adjust text accordingly.

**We adjusted the text accordingly, only mentioning bottom temperature now, see L. 78-79.**

Section 2,1,1 : The reader do not know where the lakes and drainage area (rivers) under investigation is situated. Add a map showing the global location and regional extent of each drainage area-lake system (rivers and lakes). Additionally add these system details (names, stations, etc.) where appropriate, ex. Table S1.

We have added a map (new Fig. 1) and detailed catchment maps are given by Jackson-Blake et al. 2022 as well as more background information on the case-studies. We now refer the readers to this study. See L. 110-111.

Good, but I read through the following paper and could not find the detailed catchment maps referred to in Figure 1. Add spatial catchment extent of upstream rivers that was modelled. Furthermore, using a figure already published requires permission from original creator, now since the author list below is similar to the list in this manuscript I leave it up to the editor to decide if a written permission to use the adjusted figure is required or not.

Jackson-Blake, L. A., Clayer, F., de Eyto, E., French, A. S., Frías, M. D., Mercado-Bettín, D., Moore, T., Puértolas, L., Poole, R., Rinke, K., Shikhani, M., van der Linden, L., & Marcé, R. (2022). Opportunities for seasonal forecasting to support water management outside the tropics. Hydrology and Earth System Sciences, 26(5), 1389–1406. https://doi.org/10.5194/hess-26-1389-2022

**Adding the spatial catchment extent to the map wouldn't help much for visibility, since the catchments would almost completely be hidden by the current dots. Note that the catchment maps are given in the supplementary material by Jackson-Blake et al. 2022. We now directly refer to it for the reader to see detailed catchment maps (see L. 116) and added this reference. Note also that the paper by Jackson-Blake et al. 2022 is published under the Creative Commons Attribution 4.0 License which allows anyone to use, adapt and modify the figures under the condition to give appropriate credit to the original work.**

Line 111 or 112 : add reference: SEAS5: the new ECMWF seasonal forecast system. Stephanie J. Johnson, (2019), https://doi.org/10.5194/gmd-12-1087-2019

Done

Good

Line122 to 123 : „Climate data where downloaded....". What do you mean in this sentence, ERA5 and/or SEAS5?

This sentence has been removed and we now provided much more details on ERA5 and SEAS5 data pre-processing steps, as requested by the other reviewer. See L. 139-150

Good

Line 139 ")" missing

Thank you, Corrected.

Good

Line 156 to 157 : Add details (equations and ex. RMSE) of this linear regression between in- versus outflow.

This is now described in more details L. 195-206 and in the supplementary material.

Good

Figure 2. consider showing mean of SEAS5 predictions and ERA5 at the same time (i.e. continue black lines into transition and target season).

We already show the mean of SEAS5 predictions, adding a line would overload the figure (now Fig. 3). Besides, the main concept of this figure is to illustrate the input data used to force the model. ERA5 data was used only over the warm-up period.

Good

Line 185 : RPSS looks to be missing from table 2 and table 3.
FRPSS was not considered for formal forecast verification because it doesn't allow to distinguish forecast performance for a given tercile. With these low performance lake forecasts that we have reported, this FRPSS didn't seem very useful to us. However, we still include it in Table 6 for reference.
Good

Line 235 to 238 : something is missing here, hysteresis should make linear relationship between ex. air temperature and water temperature rather bad. Describe how good these linear fits were (in appendix). And/or show with figure and improve explanation.
Pearson partial correlation coefficients (PPCC) are calculated from seasonal means, and not daily values, which likely yielded much cleaner correlation than expected from daily values. This point is made clear now see L. 321.
Good

Line 243 the reader are not familiar with the contributions of local heat fluxes at the chosen locations. Before disregarding for example cloud cover from the analysis, show the reader in numbers (or preferably figure as appendix) for each lake the seasonal heat budget contributions. I.e. uptake and emission of infrared longwave radiation, evaporation + condensation, sensible heat flux and uptake of surface downward solar radiation. Throughflow you only have the outflow (at some lakes?) since inflow temperature is missing.
This background information is now described L. 207-211 and we refer to the supplementary information L. 330 for further details.
Good

Line 250 : RMSE not consistent with RMSE/sd in Table S2. What is RMSE/sd? Use the same in text as in Table S2.
This is now clarified, these first performance measures are for the whole year (now in Table 4) while Table S2 shows measures by season. RMSE/sd is now defined in the caption of Table S2.
Good

Table 5. move description of asterisk under table and improve the site representation. Now you can not see what belongs to which system. And define the season duration.
Done.
Good

Figure 3 and 4 : missing Germany and Australia, add or explain.
Note that we precise that the four probabilistic sensitivity analyses, S-SA, W-SA, W+M0-SA and OAT-SA were only performed at the sites in Spain and Norway because of the significant resources needed to execute these hindcast experiments (see L. 296-298). This point is now repeated in the figure captions (now Fig. 4 and 5).
Good

Figure 4 : Something do not add up in your analysis. Top row for Spain – Bottom temperature, and Norway - Surface temperature appear to be to large compared to the individual season values taken together. I.e. if the impact is small most seasons, I do not see how it could be much larger on an annual basis.
We agree with you that it can appear quite surprising that the sensitivity calculated on an annual basis is much larger that over each single season. However, Norway's climate is subjected to strong seasonality which can be well captured over annual scales, but still have low correlation for a given season. To make this point clearer, we added a sentence in the figure caption (now Fig. 5). See L. 451-454.
I don't follow you reasoning here. One would expect that surface temperature (ST) in summer is sensitive to for example short wave radiation (SWR), but during a complete year this sensitivity should

be lessened by decreased importance of SWR for ST. If the seasonal signal intrude into multiple time frames (summer, autumn etc.) I would expect that the dependency is carried over from time window to time window and thereby show up for each season in your analysis.

**The grouping of several datasets showing weak correlation to a specific variable can lead to a high correlation to this variable, when taken together. The following paper shows theoretical examples of misleading correlation interpretations:**
**https://www.ncbi.nlm.nih.gov/pmc/articles/PMC5079093/.**
**For example, in Figure 5, we see that the sensitivity of Surface Temperature at the Norwegian site appears sensitive to precipitation over the whole year (panel d), However, when taking each season separately, we see much lower sensitivity. The yearly cycle of precipitation, or any other variable showing some synchronous cyclicity, and lake Surface Temperature will obviously be correlated, but by looking into the correlations at the seasonal level, we see that this correlation doesn't hold. This is why, in section 3.4, we only highlight the input variables for which sensitivity is found for specific seasons.**

Furthermore from the new figure (Fig. 5) it is now more clear what the size of the circles represent. Yet I could not find the definition for $R_2$, is it the coefficient of determination? If so, then there is an error in the figure and on line 319 since a high coefficient of determination (towards 1) show good correlation, thereby $1-R_2$ would give small and not large circles for more influential input variables. Is this so?
**It is the other way around, the R2 here is comparing two model outputs, Lake_PO and the outputs generated with one input variable that has been replaced by random data. So, when both outputs are associated with a R2 = 1, the randomized input variable had no influence on the output. Hence, "1-R2" is a measure of output sensitivity to a specific input variable. To make sure this point is clear, we added a sentence in the figure caption, see L. 459.**

Figure 5 : Why so many data gaps? Consider showing seasons where significance is worse (higher) but clearly state which significance level you trust.
This figure has been updated to show all correlation coefficients, even those that are not significant, and we now precise the significance level we trust in the figure caption (now Fig. 6) at L. 458-459.
Good, but use different color range for PPCC, try for example white color around 0.
**Thank you, we now use white color around 0 and it has significantly improved the readability of the figure.**

Line 468 : add author contributions. Who did what?
Done, see L. 626-629
Good

Additional points
It need to be more clear what is included in the lower, middle and upper terciles used throughout the manuscript. As an example it is now hard to understand Table 5, which show that your predictions can be better for lower and upper terciles compared to middle. I understand it as terciles composing a diversion of the model output so that extremes end up in the lower and upper terciles. With this understanding it now looks like from Table 5 that you are better at predicting extremes rather than the majority of events occurring in your system.
**Note that a tercile includes 33% of the whole data distribution, which is different from extremes. To make sure this point is clear, we added a sentence defining the upper, middle and lower terciles, see L. 254-256.**

Add the following from Supplement to the main manuscript

You can find all the codes and data files related to this manuscript at: https://github.com/NIVANorge/seasonal_forecasting_watexr
**Done see L. 637-638**

Figure 4: Extend y-range of plots, now boxplots are cut-off by the x-axis
**Done**

Line 239: Incorrect title number labeling

**Corrected**

**Additional edits:**

**Note that we performed some additional minor edits to improve the readability and precision of the manuscript. (See track-change version of the manuscript), as well as updates in the affiliations of the authors.**